# Phosphorylation of human glioma-associated oncogene 1 on Ser937 regulates Sonic Hedgehog signaling in medulloblastoma

Ling-Hui Zeng [1,6] ✉, Chao Tang[2,3,6], Minli Yao[2,4,6], Qiangqiang He [2,4], Meiyu Qv[1,2], Qianlei Ren[1], Yana Xu [2,4], Tingyu Shen[2,4], Weizhong Gu[3], Chengyun Xu[1,2,3], Chaochun Zou[3], Xing Ji[1,2], Ximei Wu [2,4] ✉ & Jirong Wang [5] ✉

Aberrant activation of sonic hedgehog (SHH) signaling and its effector transcriptional factor GLI1 are essential for oncogenesis of SHH-dependent medulloblastoma (MB_{SHH}) and basal cell carcinoma (BCC). Here, we show that SHH inactivates p38α (MAPK14) in a smoothened-dependent manner, conversely, p38α directly phosphorylates GLI1 on Ser937/Ser941 (human/ mouse) to induce GLI1's proteasomal degradation and negates the transcription of SHH signaling. As a result, $Gli1^{S941E}$ loss-of-function knock-in significantly reduces the incidence and severity of *smoothened-M2* transgene-induced spontaneous MB_{SHH}, whereas $Gli1^{S941A}$ gain-of-function knock-in phenocopies *Gli1* transgene in causing BCC-like proliferation in skin. Correspondingly, phospho-Ser937-GLI1, a destabilized form of GLI1, positively correlates to the overall survival rate of children with MB_{SHH}. Together, these findings indicate that SHH-induced p38α inactivation and subsequent GLI1 dephosphorylation and stabilization in controlling SHH signaling and may provide avenues for future interventions of MB_{SHH} and BCC.

Hedgehog (HH) signaling pathway is critical for embryonic development and tumorigenesis[1,2]. Mammalian HH ligands consist of sonic hedgehog (SHH), indian hedgehog (IHH), and desert hedgehog (DHH). Binding of HH ligands to receptors patched (PTCH) relieves the inhibition of smoothened (SMO), a co-receptor of HH pathway, and then allows the glioma-associated oncogene (GLI) family transcriptional factors including GLI1, GLI2, and GLI3 to induce transcription of target genes, including Hedgehog-interacting protein (*Hhip*), *Cyclin D, Cyclin*

*E*, *Ptch1* and *Gli1*[3,4]. GLI2 and GLI3 transcription factors are processed in a proteasome-dependent manner to generate a transcriptional activator or repressor in response to HH stimulation, and GLI1, not only a transcriptional factor but also a target protein of HH signaling pathway, is conventionally regarded as a transcriptional activator and dispensable for proteasome-dependent processing[5,6]. Suppressor of fused (SUFU), a cytoplasmic protein, interacts with all the three GLI proteins and serves as a repressor of GLI transcriptional factors[7,8].

[1]Key Laboratory of Novel Targets and Drug Study for Neural Repair of Zhejiang, Province, Hangzhou City University School of Medicine, Hangzhou 310015, China. [2]Department of Pharmacology, Zhejiang University School of Medicine, Hangzhou 310058, China. [3]National Clinical Research Center for Child Health, the Children's Hospital of Zhejiang University School of Medicine, Hangzhou 310053, China. [4]Department of Orthopaedics, the Affiliated Sir Run Run Shaw Hospital, Zhejiang University School of Medicine, Hangzhou 310016, China. [5]Department of Geriatrics, Zhejiang Provincial Key Lab of Geriatrics, Zhejiang Hospital, Hangzhou 310030, China. [6]These authors contributed equally: Ling-Hui Zeng, Chao Tang, Minli Yao. ✉e-mail: zenglh@zucc.edu.cn; xiwu@zju.edu.cn; wangjr@zju.edu.cn

Medulloblastoma (MB) is one of the most common cerebellar malignancies of childhood, and accounts for about 20% of all childhood brain tumors[9]. MB has four molecular groups including WNT group, SHH group (MB$_{SHH}$), group 3, and group 4, among them, MB$_{SHH}$ is the most common form and makes up approximate 30% of all MB cases[10]. Basal cell carcinoma (BCC) is the most common form of skin cancer and the most frequently occurring form of all cancers[11]. HH-related genetic alterations occur frequently in MB$_{SHH}$ (87%) and BCC (85%)[12], and loss-of-function mutations of *Ptch1* and *Sufu* and gain-of-function mutations of *Smo* and *Gli2* are typically found in MB$_{SHH}$ and BCC, all resulting in the upregulation of GLI1 levels[13]. Therefore, GLI1 is believed to be a critical effector in the incidence and severity of these two types of cancers[14]. Ablation of *Gli1* in haploinsufficient *Ptch1*[+/-] mice significantly reduces spontaneous MB$_{SHH}$ formation[15]. Due primarily to the importance of SHH signaling pathway, SMO inhibitors vismodegib and sonidegib have been approved for the treatment of BCC and are undergoing the clinical trial for pharmacotherapy of MB$_{SHH}$[16]. Despite the potency and efficacy in treatment of MB$_{SHH}$ carrying *Ptch1* or *Smo* mutations, vismodegib is not effective in MB$_{SHH}$ that develops compensatory mechanisms and restores SHH signaling to the post-receptor level, and almost has no impact on MB$_{SHH}$ with genetic alterations of *Sufu* or *Gli*[17]. Thus, identification of methods and molecules with the ability to block the signaling at the post-receptor level, especially at the GLI1 level, could be a preferable choice for the therapeutic intervention of these diseases.

In this work, we demonstrate that SHH inactivates p38α to dephosphorylate hGLI1 on Ser937 (mGli1 on Ser941), thereby stabilizing GLI1 to induce the transcriptional output of SHH signaling. As a result, loss-of-function mutation of *Gli1*[S941E] reduces the incidence and severity of the constitutively active form of Smo-induced MB$_{SHH}$, whereas, gain-of-function mutation of *Gli1*[S941A] phenocopies the *Gli1* transgene in causing BCC-like proliferation in skin. Importantly, the level of p-Ser937-GLI1 positively correlates with the overall survival of patients with MB$_{SHH}$.

## Results

### Activation of SHH signaling inactivates p38 to induce GLI1 protein

SHH signaling pathway is required for the proliferation of cerebellar granule neuron precursors (GNPs) and aberrant activation of this signaling pathway gives rise to MB$_{SHH}$[18,19]. To generate a mouse model of MB$_{SHH}$, we crossed the mice conditionally expressing constitutively active form of Smo (*Smo-M2*) with the mice expressing *Cre* recombinase under the control of human glial fibrillary acidic protein (*GFAP*) promoter. Cerebella from *GFAP-Cre;SMO-M2* pups carrying MB$_{SHH}$ and from *Smo-M2* control littermates were harvested at 2 weeks for quantitative iTRAQ LC-MS/MS phosphoproteomic analysis. A total of 7880 phosphorylation sites for 3265 phosphoproteins were identified and quantified, and proteins with a fold change greater than or equal to 2.0 and less than or equal to 0.5 with $p < 0.05$ were considered significantly differential expression. Among them, 749 differentially expressed phospho-proteins were significantly upregulated, such as ribonucleoside-diphosphate reductase subunit M2 (Rrm2), rap guanine nucleotide exchange factor 2 (Rapgef2), and formin-binding protein 1-like (Fnbp1l), and 1343 differentially expressed phospho-proteins were significantly downregulated, such as tripartite motif-containing protein 2 (Trim2), microtubule-associated protein 1 A (Map1a), and mitogen-activated protein kinase 14 (MAPK14) (Fig. 1a, b).

Since the expression of phosphorylated MAPK14, p38α, was strikingly downregulated in *GFAP-Cre;Smo-M2* cerebella, we investigated the potential role of p38 in the regulation of SHH signaling and MB$_{SHH}$. *GFAP-Cre;Smo-M2* pups at 2 weeks exhibited a robust increase in the cerebellar volume and an apparent loss of cerebellar gyri, when compared to *Smo-M2* pups (Fig. 1c). Hematoxylin-eosin (H&E) staining

of cerebellar sections showed *GFAP-Cre;Smo-M2* pups developed MB$_{SHH}$ with ectopic proliferation within the molecular layer (ML) and delayed differentiation of granule cells (Fig. 1c). Immunochemistry staining indicated p-Thr180/Tyr182-p38 (p-p38), the prerequisite for p38 activity, was predominantly expressed in the ML and minimally expressed in the granular cell layer in *Smo-M2* cerebella, whereas p-p38 expression was robustly decreased in both the compartments of *GFAP-Cre;Smo-M2* cerebella. In contrast, Gli1 expression was almost undetectable in *Smo-M2* cerebella but was robustly and diffusely distributed throughout the *GFAP-Cre;Smo-M2* cerebella (Fig. 1c). Likewise, western blot analyses revealed *GFAP-Cre;Smo-M2* cerebella showed a dramatic decrease and increase in p-p38 and Gli1 levels, as compared to *Smo-M2* cerebella, respectively (Fig. 1d). The inactivation of p38 and the increase in GLI1 protein were further confirmed by immunohistochemistry analysis in clinical MB$_{SHH}$ sections, which showed MB$_{SHH}$ had much less and more p-p38- and GLI1-positive cells than non-WNT/non-SHH MB (group 3 or 4 MB), respectively (Fig. 1e). Thus, activation of SHH signaling substantially induced MB$_{SHH}$ and concurrently inactivated p38 in cerebella.

We next examined the phosphorylation levels of three main subfamilies of MAPKs in the SHH-responsive embryonic fibroblasts, C3H10T1/2 cells, and primary mouse embryonic fibroblasts (MEFs). Bioactive SHH recombinant protein, N-SHH, or Smo agonist, purmorphamine, time-dependently reduced the phosphorylation level of p38 but not of extracellular regulated protein kinase 1/2 (ERK1/2) and c-Jun N-terminal kinase (JNK) in C3H10T1/2 cells or MEFs, respectively (Fig. 1f and Supplementary Fig. 1a). In contrast, genetic ablation of *Smo* in MEFs markedly induced the basal levels of p-p38 and almost completely restored the negative effect of N-SHH on p-p38 (Supplementary Fig. 1b). Thus, SHH inactivated p38 in a Smo-dependent manner.

To explore the potential role of p38 in the regulation of SHH signaling, we treated C3H10T1/2 cells with a specific p38 inhibitor, SB203580. Interestingly, SB203580 induced the Gli1 protein but not the full-length of Gli2 (Gli2F) and Gli3 (Gli3F) proteins or their truncation forms including N-terminally truncated Gli2 (Gli2 activator, Gli2A) and C-terminally truncated Gli3 (Gli3 repressor, Gli3R) in a time-dependent manner, and maximized the induction of Gli1 protein at 1 h post-treatment (Fig. 1g). In addition, SB203580 significantly induced the *Gli-luciferase* reporter gene expression in a dose-dependent manner in C3H10T1/2 cell expressing *Gli1* (Supplementary Fig. 1c). To determine whether p38 participated in the post-translational modifications of Gli1 protein or the transcription of *Gli1* gene, we performed quantitative RT-PCR (qRT-PCR) in C3H10T1/2 cells treated with N-SHH or SB203580 for different times. N-SHH and SB203580 began to induce the *Gli1* mRNA expression at 3 and 9 h post-treatments and to induce the *Ptch1* mRNA expression at 12 and 12 h post-treatments, respectively (Supplementary Fig. 1d, e). SB203580 treatment induced the expression of Gli1 protein earlier than *Gli1* mRNA, suggesting that p38 mediated the post-translational regulation of Gli1.

The specific role of p38 in the regulation of Gli1 protein was further confirmed by western blot analyses. Treatment with SB203580 but not PD98059, an ERK1/2 inhibitor, or SP600125, a JNK inhibitor, markedly increased the Gli1 protein (Fig. 1h). Overexpression of MAPK14 (p38α) but not MAPK11 (p38β), MAPK12 (p38γ) or MAPK13 (p38δ) robustly negated the Gli1 protein (Fig. 1I, j), and significantly reduced not only the *Gli-luciferase* reporter gene expression but also the nuclear and cytosolic fractions of Gli1 in C3H10T1/2 cells (Supplementary Fig. 2a−c). Conversely, treatment with a selective p38α/β inhibitor, losmapimod, robustly induced the Gli1 protein (Fig. 1k), and significantly increased not only the *Gli-luciferase* reporter gene expression but also the nuclear and cytosolic fractions of Gli1 in C3H10T1/2 cells (Supplementary Fig. 2d, e). Thus, p38α but not p38β, p38γ, and p38δ negated the Gli1 protein to reduce the transcriptional output of SHH signaling.

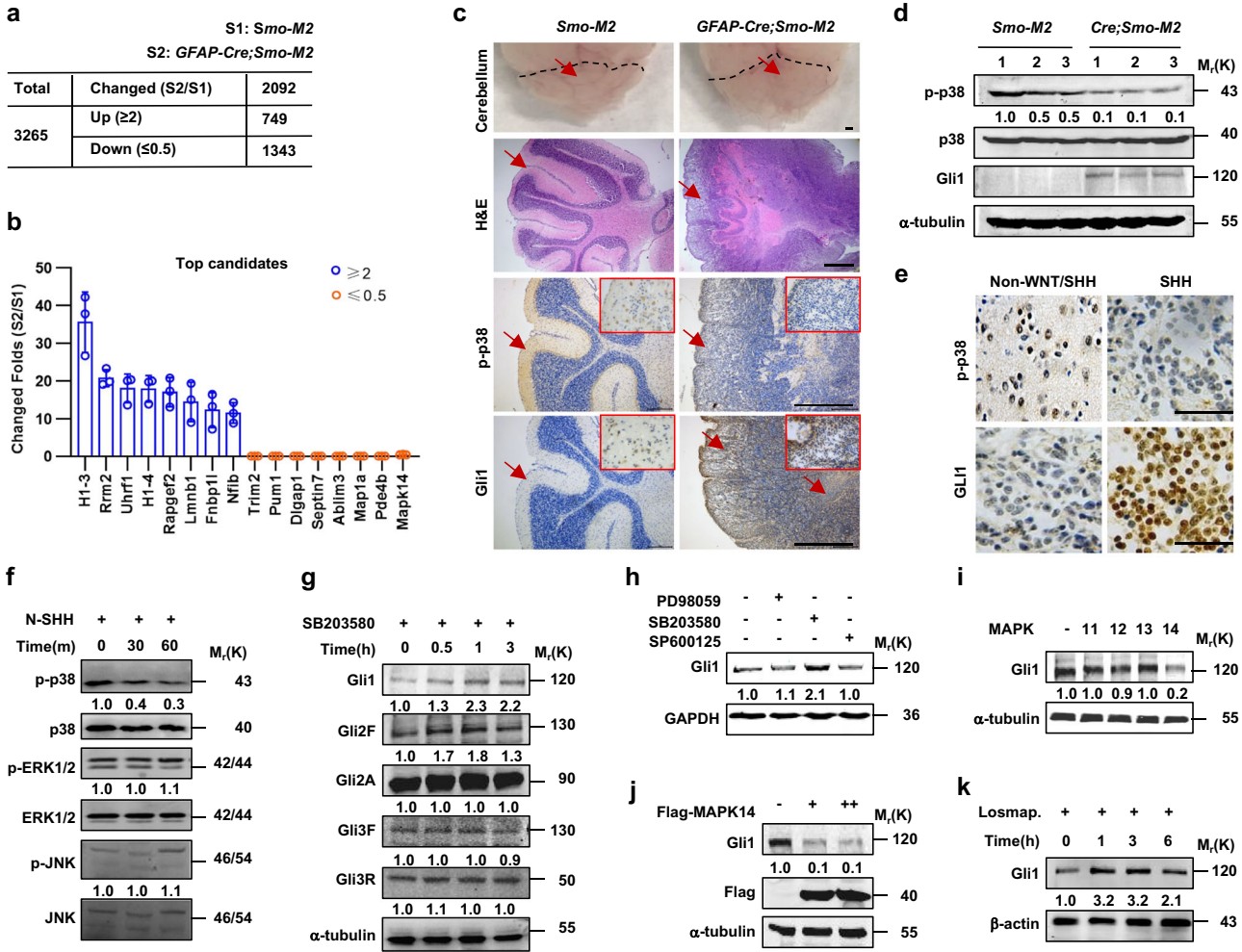

**Fig. 1 | Activation of SHH signaling inactivates p38 to induce Gli1 protein.**
**a**, **b** Cerebella from *GFAP-Cre;Smo-M2* pups bearing spontaneous MB$_{SHH}$ and from *Smo-M2* control littermates at 2 weeks of age were harvested for quantitative iTRAQ LC-MS/MS phosphoproteomic analysis (*n* = 3 mice per group). Numbers of the differentially expressed phosphoprotein (**a**) and top candidates among them were listed (**b**). **c** H&E and immunohistochemistry staining of cerebella from above mice. Bar 200 µm. **d** Western blot analyses of cerebella from above mice (*n* = 3 mice per group). **e** Immunohistochemistry analyses for p-p38 and GLI1 in paraffin-embedded sections of non-WNT/non-SHH MB and MB$_{SHH}$. Bar, 20 µm. **f–h** Western blot analyses in C3H10T1/2 cells treated with N-SHH (100 ng/ml), SB203580 (10 µM), PD98059 (10 µM), or SP600125 (10 µM) for the indicated times or 6 h. **i–k** Western blot analyses in C3H10T1/2 cells after transfection with the MAPK11, MAPK12, MAPK13 or MAPK14 construct for 48 h or after treatment with losmapimod at 10 µM for the indicated times. Data were presented as mean ± sd, two-tailed Student's *t* test for (**b**). A representative example of three replicates is shown for **c–k**. Source data are provided as a Source Data file.

## P38α phosphorylates GLI1 on Ser937 (mouse Ser941) to destabilize GLI1

To explore the physical interaction between p38 and GLI1, we performed co-immunoprecipitation experiments in HEK293T (293T) cells transiently expressing Flag-MAPK14 or HA-GLI1. Immunocomplexes precipitated with a Flag or HA antibody contained abundant endogenous GLI1 or p-p38 in addition to Flag-MAPK14 or HA-GLI1 as expected, respectively (Supplementary Fig. 3a, b). To investigate the potential role of p38 in phosphorylating GLI1, we treated the 293T cells expressing HA-GLI1 with losmapimod and performed co-immunoprecipitation with an HA antibody. Western blot analyses using a phospho-Ser/Thr antibody showed losmapimod time-dependently decreased the phospho-Ser/Thr levels of HA-GLI1 (Fig. 2a). Thus, p38α interacted with GLI1 and potentially phosphorylated GLI1.

Prediction of phosphorylation sites on GLI1 was performed by using a BioGPS database (biogps.org), and revealed seven potential p38 phosphorylation sites i.e., Ser70, Ser569, Ser686, Ser885, Ser927, Ser937, and Ser968. To assess their potential importance, we expressed GLI1 variants harboring mutations at the consensus serine residues

(Ser to Ala) individually (S70A, S569A, S686A, S885A, S927A, S937A, and S968A) and evaluated their ability to mediate *Gli-luciferase* reporter gene expression. Overexpression of wild-type GLI1 (WT) increased the reporter gene expression by approximately 3.4-fold. Though the S70A, S569A, S686A, S885A, S927A, and S968A variants behaved essentially the same as WT, the S937A variant caused an approximately 2.4-fold increase over the WT (Fig. 2b). In addition, high-resolution mass spectrometry (MS) analysis for GLI1 purified from Daoy cells, human medulloblastoma cells, revealed six endogenous potential phosphorylation sites, including Ser70, Ser569, Ser927, Ser937, Thr601 and Thr1074 (Supplementary Fig. 4). Sequence alignment revealed a conserved phosphorylation motif at S937 (mouse S941) for the GLI1 protein (Fig. 2c). Interestingly, GLI1(S937A) caused a significant increase in the basal level of GLI1 protein (Fig. 2d, e). Overexpression of MAPK14 largely reduced GLI1(WT) protein and GLI1(WT)-induced luciferase reporter gene expression but had no effect on GLI1(S937A) protein and GLI1(S937A)-induced luciferase expression, in contrast, treatment with losmapimod robustly potentiated GLI1(WT) protein and GLI1(WT)-induced luciferase expression but had no effect on GLI1(S937A) protein and GLI1(S937A)-induced

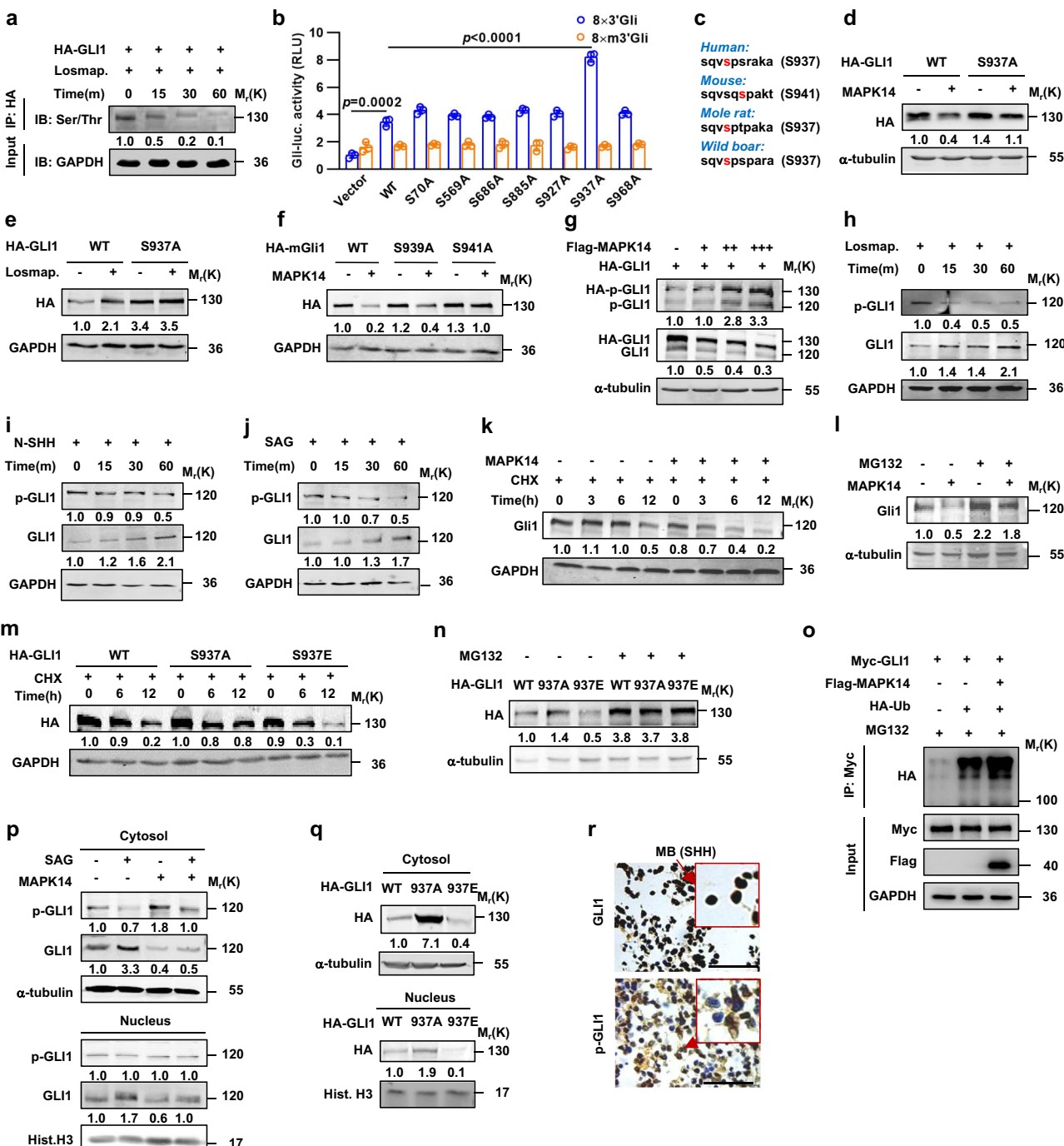

**Fig. 2 | p38α phosphorylates GLI1 on Ser937/Ser941 to destabilize GLI1. a** Co-immunoprecipitation and western blot analyses in 293T cells transfected with HA-GLI1 construct and treated with losmapimod at 10 μM for the indicated times. Western blot analyses were performed using anti-Ser/Thr and GAPDH antibodies. **b** Gli-luciferase reporter assays in 293T cells transiently co-transfected with Gli-luciferase/Gli(mutant)-luciferase and Renilla-luciferase reporter constructs and GLI1 variants (*n* = 3 independent experiments). **c** The conserved Ser937/Ser941 sites of GLI1 among species. **d**–**f** Western blot analyses in 293T cells transfected with HA-GLI1/HA-mGli1 or their S937A/S939A/S941A variants in combination with or without MAPK14 and treated with or without losmapimpod at 10 μM for 6 h. **g**–**j** Western blot analyses of phospho-S937-GLI1 (p-GLI1) and GLI1 in 293T cells transfected with the increasing amounts of MAPK14 or treated with losmapimpod

at 10 μM, N-SHH at 100 ng/ml or SAG at 2.5 μM for the indicated times. **k**–**n** Western blot analyses in C3H10T1/2 cells transfected with or without MAPK14 or HA-GLI1 variants and treated with cycloheximide at 2 μg/ml for the indicated times or MG132 at 10 μM for 12 h. **o** C3H10T1/2 cells expressing Myc-GLI1, HA-ubiquitin, and Flag-MAPK14 were immunoprecipitated with anti-Myc for western blot analyses. **p**, **q** Western blot analyses of cytosolic versus nuclear fractions of the indicated proteins in C3H10T1/2 cells transfected with MAPK14 and treated with SAG at 2.5 μM for 6 h or transfected with HA-GLI1 variant alone. **r** Immunohistochemistry staining for phospho-S937-GLI1 in paraffin-embedded children MB_SHH sections. Bar, 20 μm. Data were presented as mean ± sd, two-tailed Student's *t* test for (**b**). A representative example of three replicates is shown for (**a**) and (**d**–**r**). Source data are provided as a Source Data file.

luciferase expression (Fig. 2d, e and Supplementary Fig. 3c, d). Notably, mouse Gli1(S941A) but not Gli1(S939A) behaved essentially the same as human GLI1(S937A) in mediating MAPK14's effect (Fig. 2f). Thus, p38α negated the GLI1 protein possibly by phosphorylating hGLI1/mGli1 on S937/S941 site.

To further determine the role of p38α in phosphorylation of GLI1 on S937, we developed a phospho-S937-GLI1 (p-GLI1) polyclonal antibody by repeatedly immunizing rabbits with a peptide of SQV(pS) PSRAKA. The specificity of p-GLI1 was confirmed by a neutralizing polypeptide, which completely diminished p-GLI1-derived immunoreactive bands, and by GLI1-shRNA, which significantly decreased the expression of GLI1 as well as p-GLI1 (Supplementary Fig. 3e, f). Overexpression of the increasing contents of MAPK14 dose-dependently induced and reduced endogenous and exogenous p-Gli1 and GLI1 levels in 293T cells transfected with Flag-MAPK14 and HA-GLI1, respectively (Fig. 2g). As expected, losmapimod treatment time-dependently decreased and increased p-GLI1 and GLI1 protein levels in 293T cells, respectively (Fig. 2h). The role of SHH signaling in the regulation of p-GLI1 was further investigated in 293T cells. Treatment with either N-SHH or smoothened agonist (SAG) significantly decreased and increased p-GLI1 and GLI1 in a time-dependent manner, respectively (Fig. 2I, j). Thus, p38α phosphorylated GLI1 on Ser937 to negate the GLI1 protein and the transcriptional output of SHH signaling.

Overexpression of MAPK14 or treatment with losmapimod induced or reduced p-GLI1 levels and concurrently decreased or increased GLI1 levels, respectively, we therefore hypothesized that p-GLI1 was a destabilized form of GLI1 and facilitated to proteasomal degradation. To test this, we performed cycloheximide (CHX) chase and proteasomal degradation assays. CHX chase assay indicated overexpression of MAPK14 significantly shortened the half-life of endogenous Gli1 protein in C3H10T1/2 cells, whereas the proteasomal inhibitor MG132 markedly restored the MAPK14's effect (Fig. 2k, l). We also mutated this Ser937 residue to the phosphomimetic glutamate (S937E) to evaluate the potential importance of GLI1 phosphorylation at Ser937. GLI1(S937A) and GLI1(S937E) exhibited a much longer and shorter half-life than GLI1(WT) in the presence of CHX, respectively, whereas MG132 consistently restored the GLI1 levels of GLI1(S937A) and GLI1(S937E) (Fig. 2m, n). Moreover, overexpression of MAPK14 significantly increased the GLI1 ubiquitination in 293T cells transfected with Myc-GLI1 and HA-ubiquitin (Fig. 2o), and SAG and MAPK14 robustly increased and decreased the cytosolic and nuclear fractions of GLI1, decreased and increased the cytosolic p-GLI1, but had no effect on nuclear p-GLI1, respectively (Fig. 2p and Supplementary Fig. 5). Whereas GLI1(S937A) and GLI1(S937E) significantly increased and decreased the cytosolic and nuclear fractions of GLI1, respectively (Fig. 2q and Supplementary Fig. 5). Likewise, immunohistochemistry analysis of clinical MB_SHH sections indicated GLI1 and p-GLI1 were abundantly expressed in the nuclei and cytosol of tumor cells, respectively (Fig. 2r). Taken together, phosphorylation of GLI1 on S937 by p38α destabilized GLI1, resulting in the decrease in nuclear accumulation of GLI1 and transcriptional output of SHH signaling as well.

### *Gli1^S941A* knock-in causes the BCC-like proliferation in the epidermis of mice

To determine the physiological relevance of p38α-mediated phosphorylation on Ser937 (mouse Ser941) of GLI1, we generated Gli1(S941A) and Gli1(S941E) knock-in mice by using CRISPR/Cas9 technique. Western blot analysis revealed that heterozygous *Gli1(S941A)^+/-* and homozygous *Gli1(S941A)^+/+* pups at 2 weeks exhibited a dose-dependent increase in Gli1 protein in whole cerebella and their cytosolic fractions, as compared to wild-type (WT) littermates, whereas heterozygous *Gli11(S941E)^+/-* and homozygous *Gli1(S941E)^+/+* pups at 2 weeks showed a dose-dependent decrease in Gli1 protein in whole cerebella and their cytosolic fractions, as

compared to WT littermates (Fig. 3a). Moreover, qRT-PCR analysis showed *Gli1(S941A)^+/+* or *Gli1(S941E)^+/+* cerebella exhibited a robust increase or decrease in *Ptch1* mRNA levels, respectively (Fig. 3b), and co-immunoprecipitation experiments showed immunocomplexes precipitated from *Gli1(S941A)^+/+* or *Gli1(S941E)^+/+* cerebella by using a Gli1 antibody contained no p-p38 but the same level of Gli1, as compared to those from WT cerebella, respectively (Fig. 3c). Thus, in line with the in vitro findings, Gli1(S941A) or Gli1(S941E) knock-in abolished the p38α-mediated phosphorylation of Gli1 on Ser941 and stabilized or destabilized Gli1 in regulating the transcriptional output of SHH signaling, respectively.

*Gli1(S941A)^+/+* cerebella showed an apparent increase in the cerebellar weight when compared to wild-type *Gli1(S941A)^-/-* cerebella (Fig. 3d). Histological analysis revealed *Gli1(S941A)^+/+* mice did not develop MB in the cerebella up to 6 months of age, and their cerebella were morphologically similar to those of *Gli1(S941A)^-/-* mice, in addition to the thickened and swelling ML (Fig. 3d, e). To confirm the phenotypes, we performed immunofluorescence staining of cerebella at P14 by using antibodies against Pax6, NeuN, and calbindin, the markers of GNPs, post-mitotic granule neurons, and Purkinje cells (PCs), respectively. Immunofluorescence staining of Pax6, NeuN, and calbindin revealed no significant developmental abnormality in *Gli1(S941A)^+/+* pups, as compared to in *Gli1(S941A)^-/-* pups (Fig. 3f). Notably, *Gli1(S941A)^+/+* mice gave rise to a large pigmentation area with nevus-like nodules in the back skin at 4 months of age, which developed BCC-like proliferation (Fig. 3g). H&E staining exhibited *Gli1(S941A)^+/+* pups at 2 weeks of age caused the robust thickness in the epidermis and the marked increase in Gli1- and Ki67-positive areas, as compared to the age- and sex-matched *Gli1(S941A)^-/-* pups (Fig. 3h, i). On the other hand, immunohistochemistry analysis revealed that *Gli1(S941A)^+/+* skins had the extreme hyperplasia in the keratin 5 (K5)-expressing basal cell layer, slight thickness in the keratin 10 (K10)-expressing suprabasal terminally differentiating layer, and large amount of keratin 17 (K17)-expressing hyperproliferative cells (Fig. 3j, k), resembling human superficial BCC. Thus, *Gli1(S941A)* gain-of-function knock-in phenocopied *Gli1* transgene in causing BCC-like proliferation in epidermis.

### *Gli1^S941E* knock-in reduces the incidence and severity of MB in mice

To determine whether Gli1(S941E) knock-in is sufficient to reduce the incidence and severity of MB_SHH, we generated *GFAP-Cre;Smo-M2;Gli1(S941E)^+/+* pups and age- and sex-matched control *GFAP-Cre;Smo-M2* and *Smo-M2* littermates and harvested the cerebella at different stages. Western blot and qRT-PCR analyses indicated both Gli1 protein and *Ptch1* mRNA levels were robustly upregulated in *GFAP-Cre;Smo-M2* cerebella at 2 weeks of age, however, *Gli1(S941E)^+/+* or *GFAP-Cre;Smo-M2;Gli1(S941E)^+/+* cerebella at this age exhibited a significant decrease in both Gli1 protein and *Ptch1* mRNA levels, as compared to *Smo-M2* or *GFAP-Cre;Smo-M2* cerebella, respectively (Fig. 4a, b). Analysis of gross appearance indicated *GFAP-Cre;Smo-M2* but not *Smo-M2* or *Gli1(S941E)^+/+* pups at 2-weeks of age gave rise to the enlarged cerebella with the decreased gyri, whereas *GFAP-Cre;Smo-M2;Gli1(S941E)^+/+* pups developed some normally sized and some enlarged cerebella (Fig. 4c). H&E staining of cerebellar sections showed *GFAP-Cre;Smo-M2* (100%) but not *Smo-M2* (0%) or *Gli1(S941E)^+/+* (0%) pups at 2-weeks of age developed the severe MB_SHH in cerebella without exception, whereas *GFAP-Cre;Smo-M2;Gli1(S941E)^+/+* pups gave rise to the normal cerebella (46.2%) or the cerebella with mild to moderate MB_SHH (53.8%) (Fig. 4c, d). Moreover, *GFAP-Cre;Smo-M2* pups showed a significant increase or decrease in the cerebellar weights or in the number of cerebellar lobules, as compared to *Smo-M2* control pups, respectively, however, *GFAP-Cre;Smo-M2;Gli1(S941E)^+/+* pups with MB_SHH indicated a significant decrease or increase in the cerebellar weights or in the number of cerebellar lobules, as compared to *GFAP-Cre;Smo-M2* pups

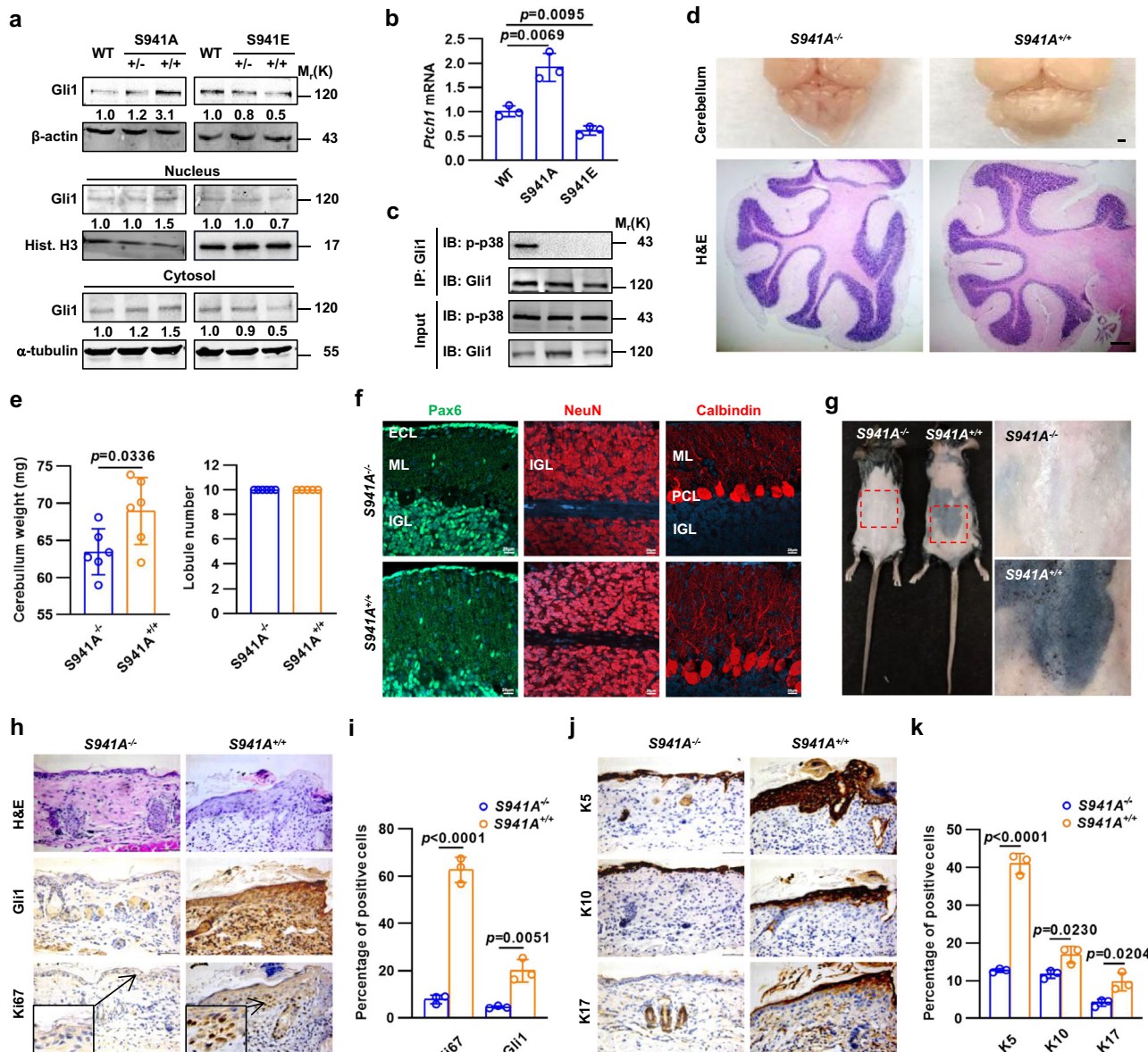

**Fig. 3 | Gli1(S941A) knock-in causes BCC-like proliferation in epidermis.**
**a** Western blot analyses of cytosolic versus nuclear Gli1 in cerebella of wild-type (WT), *Gli1(S941A)*⁺/⁻, *Gli1(S941A)*⁺/⁺, *Gli1(S941E)*⁺/⁻ or *Gli1(S941E)*⁺/⁺ pups at 2 weeks of age. **b**, **c** Quantitative RT-PCR analysis for *Ptch1* mRNA and co-immunoprecipitation experiments in cerebella of WT, *Gli1(S941A)*⁺/⁺ or *Gli1(S941E)*⁺/⁺ pups at 2 weeks of age (*n* = 3 mice per group). **d** Gross appearance of the cerebella and H&E staining for cerebellar sections from *Gli1(S941A)*⁻/⁻ and *Gli1(S941A)*⁺/⁺ pups at 2 weeks of age. Bar, 200 μm. **e** Quantification of cerebellar weights and lobule number in *Gli1(S941A)*⁻/⁻ and *Gli1(S941A)*⁺/⁺ pups (*n* = 6 mice per group). **f** Anti-Pax6, anti-NeuN and antin-

Calbindin labeling in sagittal sections of the cerebellar vermis of *Gli1(S941A)*⁻/⁻ and *Gli1(S941A)*⁺/⁺ pups. Bar, 20 μm. **g** Gross appearance of skin of *Gli1(S941A)*⁻/⁻ and *Gli1(S941A)*⁺/⁺ mice at 4 months of age. **h**–**k** H&E staining and immunohistochemistry analyses and quantification for Gli1, Ki67, K5, K10, and K17 in skin sections of *Gli1(S941A)*⁻/⁻ and *Gli1(S941A)*⁺/⁺ pups at 2 weeks of age (*n* = 3 mice per group), Bar, 50 μm. Data were presented as mean ± sd, two-tailed Student's *t* test for (**b**, **e**, **i** and **k**). A representative example of three replicates is shown for (**a**, **c**, **f** and **g**). Source data are provided as a Source Data file.

with MB_SHH (Fig. 4e). Finally, NeuN immunostaining revealed that a large number of NeuN⁺ granule cells were dispersedly localized in the disarranged ML of *GFAP-Cre;Smo-M2* cerebella that were restricted to granule neurons within the IGL in *Smo-M2* or *Gli1(S941E)*⁺/⁺ cerebella, whereas ectopically expressed NeuN⁺ cells were significantly reduced as evidenced by the fact that most of the NeuN⁺ cells were restrictedly localized in the relatively well-arranged IGL of *GFAP-Cre;Smo-M2;Gli1(S941E)*⁺/⁺ cerebella (Fig. 4f, g). In the *GFAP-Cre;Smo-M2* cerebella, calbindin⁺ PCs and brain lipid-binding protein⁺ (BLBP⁺) Bergmann glial cells (BGs) were significantly increased and dispersed, whereas in the *GFAP-Cre;Smo-M2;Gli1(S941E)*⁺/⁺ cerebella these cells were largely restored to normal number and distribution (Fig. 4f, g).

Thus, *Gli1(S941E)* knock-in decreased the Gli1 protein and transcriptional output of SHH signaling to attenuate the incidence and severity of MB_SHH in mice.

Analyses of the cell apoptosis showed a similar level of TUNEL staining in the molecular and granular layers of *Smo-M2*, *Gli1(S941E)*⁺/⁺ or *GFAP-Cre;Smo-M2* cerebella, but a significant increase in TUNEL staining in both compartments of *GFAP-Cre;Smo-M2;Gli1(S941E)*⁺/⁺ cerebella (Fig. 4h, k). Moreover *GFAP-Cre;Smo-M2* cerebella exhibited an apparent increase in Ki67- and Gli1-postitive staining, as compared to *Smo-M2* cerebella, and *Gli1(S941E)*⁺/⁺ cerebella had a similar Ki67- and Gli1-postitive staining to *Smo-M2* cerebella, whereas *GFAP-Cre;Smo-M2;Gli1(S941E)*⁺/⁺ cerebella showed a marked decrease in Ki67- and Gli1-

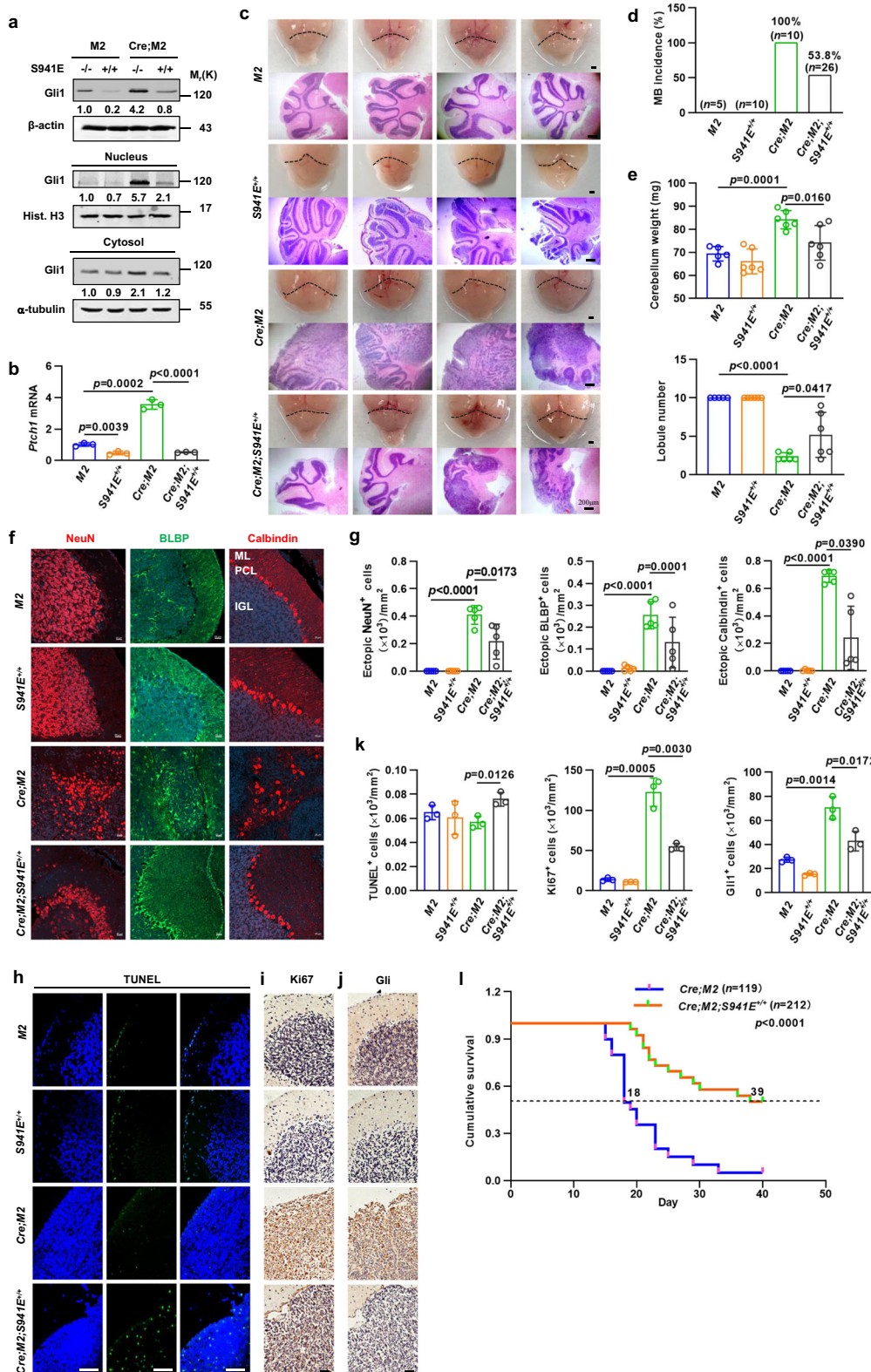

positive staining, as compared to *GFAP-Cre;Smo-M2* cerebella (Fig. 4i–k). Finally, we harvested *GFAP-Cre;Smo-M2* pups with MB$_{SHH}$ and *GFAP-Cre;Smo-M2;Gli1(S941E)^{+/+}* pups with or without MB$_{SHH}$ for survival curve analysis. Log-rank analysis revealed that the overall survival of *GFAP-Cre;Smo-M2* pups (n = 119) was significantly shorter than that of *GFAP-Cre;Smo-M2;Gli1(S941E)^{+/+}* pups (n = 212), and the median survival of *GFAP-Cre;Smo-M2* and *GFAP-Cre;Smo-M2;Gli1(S941E)^{+/+}* pups was 18 and 39 postnatal days, respectively (p < 0.001) (Fig. 4l). Thus, *Gli1(S941E)*

knock-in was indeed sufficient to reduce the incidence and severity of MB$_{SHH}$ in *GFAP-Cre;Smo-M2* pups.

To determine the potential relevance between p38α-phosphorylating GLI1 on Ser937 and tumorigenesis, we performed in vitro experiments using Daoy cells. Treatment with SAG, significantly increased the proliferation rate of Daoy cells, overexpression of MAPK14 or GLI1(S937E) robustly negated the SAG's effects, in contrast, overexpression of GLI1(S937A) significantly attenuated the

**Fig. 4 | Gli1(S941E) knock-in reduces the incidence and severity of MB_SHH in** *GFAP-Cre;Smo-M2* **pups. a, b** Western blot analyses of cytosolic versus nuclear Gli1 and qRT-PCR analysis for *Ptch1* mRNA in cerebella of *Smo-M2, Gli1(S941E)*[+/+], *GFAP-Cre;Smo-M2*, and *GFAP-Cre;Smo-M2;Gli1(S941E)*[+/+] pups at 2 weeks of age (*n* = 3 mice per group). **c** Gross appearance of cerebella and H&E staining for cerebellar sections from the above pups at 2 weeks of age. Bar, 200 μm. **d** The incidence of MB_SHH in the above pups at 2 weeks of age. **e** Quantification of cerebellar weights and lobule number of sagittal sections of the cerebellar vermis in the above pups (*n* = 5 mice for *Smo-M2*, and *n* = 6 mice for others). **f, g** Anti-Calbindin, anti-BLBP, and anti-NeuN labeling in sagittal sections of the vermis and the quantification (*n* = 5 mice

per group), Bar, 20 μm. **h–k** Representative images for TUNEL staining and immunohistochemistry staining for Ki-67 and Gli1 in cerebellar sections from the above pups at 2 weeks of age (*n* = 3 mice per group). Bar, 20 μm. **l** Overall survival rates of *GFAP-Cre;Smo-M2* pups with MB_SHH (*n* = 119 mice) and *GFAP-Cre;Smo-M2;Gli1(S941E)*[+/+] (*n* = 212 mice) pups with or without pathologically proven MB_SHH. Median survival times are shown. Data were presented as mean ± sd, two-tailed Student's *t* test for (**b, e, g, k**), Chi-squared test for (**d**), Log-rank analysis for (**l**). A representative example of three replicates is shown for (**a**). Source data are provided as a Source Data file.

inhibitory effect of cyclopamine, a SMO antagonist, on Daoy cell proliferation (Supplementary Fig. 6a). Wound healing and transwell assays were performed to determine the migration and invasion of tumor cells. Overexpression of MAPK14 significantly decreased the SAG-induced migration and invasion of Daoy cells (Supplementary Fig. 6b, c), however, overexpression of GLI1(S937A) significantly enhanced the migration and invasion of tumor cells without affecting the cyclopamine's effects (Supplementary Fig. 6d). Thus, p38α-mediated phosphorylation on Ser937 of GLI1 was involved in the tumorigenesis of medulloblastoma and the sensitivity to chemotherapy of SMO antagonists.

To determine the pathological relevance between p38α-phosphorylating GLI1 on Ser937 and severity of human MB, we performed a retrospective case-control study. Forty-six MB children who received surgery at the National Clinical Research Center for Child Health, Children's Hospital of Zhejiang University School of Medicine were recruited, and 20 and 26 of them were pathologically diagnosed with Group 3 or 4 MB and MB_SHH, respectively. Immunohistochemistry staining was performed on MB sections to semi-quantify the levels of p-GLI1 and p-p38. Cytosolic p-GLI1 levels from sections of MB_SHH were substantially lower than those from sections of non-WNT/non-SHH MB (Group 3 or 4 MB), though their nuclear levels of p-GLI1 displayed no significant difference (Fig. 5a, b). Phospho-p38 levels were found to be robustly decreased in MB_SHH versus Group 3 or 4 MB (Fig. 5a, c). In addition, correlation analysis indicated cytosolic p-GLI1 levels positively correlated with p-p38 levels in MB_SHH and the $R^2$ value was 0.5656 (*p* < 0.0001) (Fig. 5d). To investigate whether the cytosolic p-GLI1 level was associated with the prognosis of MB_SHH, we calculated the overall survival rates for 13 MB_SHH with high expression of p-GLI1 and 13 MB_SHH with low expression of p-GLI1. Log-rank analysis showed that the overall survival of children with high cytosolic p-GLI1 expression was significantly longer than that of children with low cytosolic p-GLI1 expression, and the median survival times for children with high and low cytosolic p-GLI1 expression were 37.0 and 19.0 months, respectively (*p* < 0.001) (Fig. 5e). However, there was no significant difference in overall survival between children with high p-p38 expression and those with low p-p38 expression (Fig. 5f). Thus, the cytosolic p-Ser937-GLI1 level may help to predict the prognosis of MB_SHH.

## Discussion

By using biochemical, genetic and clinical approaches, we uncover a signaling cascade that operates in conjunction with Gli1 phosphorylation on Ser937/Ser941 to regulate SHH signaling. In the absence of SHH, p38α phosphorylates GLI1 on Ser937/Ser941, and thereby promoting proteasomal degradation of GLI1, resulting in the inactivation of SHH signaling; in the presence of SHH, the SMO-mediated inhibition of p38α decreases the phosphorylation of GLI1, resulting in the stabilization of GLI1 and transcriptional output of SHH signaling. As a result, p-Ser937-GLI1 positively correlates with the overall survival of children with MB_SHH; loss-of-function of Gli1(S941) mutation reduces the incidence and severity of constitutively active form SMO-induced mouse MB_SHH, whereas gain-of-function of Gli1(S941) mutation phenocopies *Gli1* transgene in causing BBC-like proliferation in mouse skin. Our

results are not only consistent with previous findings that GLI1 phosphorylation at certain sites regulates its stability, localization, and transcriptional activity[20–28], but also identify the Ser937/Ser941 phosphorylation as a determinant mechanism controlling SHH signaling as well as the incidence and severity of MB_SHH and BCC.

A C-terminal motif of Gli1, DSGVEM, has been identified to stabilize the Gli1 protein and to rapidly accelerate the tumor formation[29]. AMP-activated protein kinase (AMPK) phosphorylates Ser102-, Ser408-, and Thr1074-GLI1, leading to its ubiquitinated degradation via β-TrCP[30,31], and the proteasome co-activator TRIM16 regulates GLI1 intracellular levels by directly mediating GLI1 ubiquitination degradation[32]. Mitogen-activated protein kinase (MAPK) kinase kinase 1/2 (MEKK1/2), the upstream activators of ERK1/2 and JNK1/2/3, promotes phosphorylation of GLI1 to inhibit its activity[33], and MEKK3, an upstream activator of p38, phosphorylates the GLI1 on Ser201, Ser204, Ser243, Ser968, Thr1074 and Ser1078 sites and consequently affects the GLI1 protein stability and ability to bind with SuFu and DNA[34]. Although p38α phosphorylates GLI1 on Ser937/Ser941 sites distinct from the sites phosphorylated by MEKK1/2/3, Ser937/Ser941 sites of GLI1 did locate at MEKK1 phosphorylation cluster between Se461 and Thr1014[35]. However, the exact proteasomal complex involved in p-Ser937/Ser941-GLI1 degradation remains unknown and worthy of further study. In addition, p38 is abundantly expressed in the primary cilium[36], we logically propose that phosphorylation of GLI1 on Ser937/Ser941 by p38 occurs in this signaling compartment for SHH pathway.

To date, four isoforms of the p38 family have been identified: p38α, p38β, p38δ, and p38γ. p38α and p38β are widespread in various tissues, whereas p38γ or p38δ is restrictedly expressed in skeletal muscle cells or glandular tissue, respectively[37]. p38 phosphorylates a myriad of substrates including transcription factors and kinases and elevated p38α activity occurs in colorectal cancer, mammary carcinomas, follicular lymphoma, glioma, head and neck squamous cell carcinoma, lung cancer, and thyroid cancer[38]. Activation of Shh signaling increased phosphorylation of p38, and inhibition of p38 significantly attenuated Shh-dependent upregulation of Gli1 in astrocytes[39], whereas phospho-p38 levels in the joint tissue of collagen-induced arthritis was decreased after blockade of SHH signaling[40]. However, a previous study on children MB_SHH-derived xenografts indicated p38 kinase MAP2K3/MKK3 and the p38 effector MAPKAPK2/MK2 were significantly upregulated, as compared to Group 3 or 4 MB xenografts[41], and a previous study demonstrated the level of p38α increased significantly during the proliferation of cerebellar granule cell precursors, and inhibition of SHH signaling or granule cell precursor proliferation decreased p38α levels[42]. These findings appear to be at odds with our study that p38 is inactivated in children and mouse MB_SHH. The discrepancy could indicate that the role of p38 in SHH signaling depends on the duration, degree, and/or cellular location of subtype activation. In this regard, the exact role of p38α in SHH signaling in vivo remains to be further investigated, and it would be of interest in future studies to complete the characterization of p38α inactivation mice and to determine the role of p38α in cell proliferation and tumorigenesis of MB_SHH.

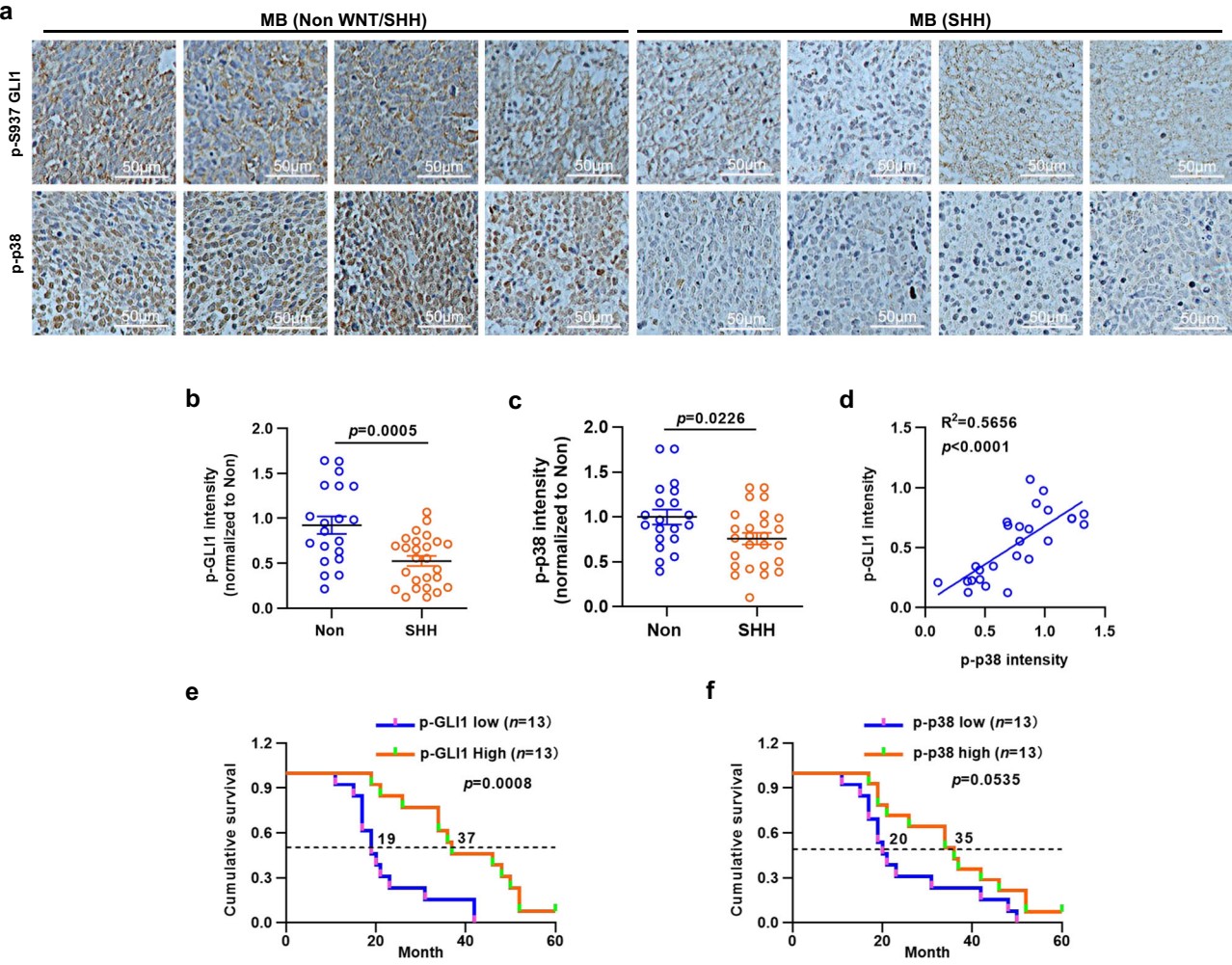

**Fig. 5 | The correlation between p-S937-GLI1/p-p38 levels and overall survival rates of children with MB$_{SHH}$. a** Representative immunohistochemistry images showing clinical non-Wnt/non-SHH and MB$_{SHH}$ samples stained with indicated antibodies. Bar, 50 μm. **b**, **c** Semi-quantification of p-S937-GLI1 and p-p38 in clinical non-Wnt/non-SHH and MB$_{SHH}$ sections (each dot represents one clinical specimen, data were pooled from ≥3 experiments, $n = 20$ for non-Wnt/non-SHH and $n = 26$ for MB$_{SHH}$). **d** Linear regression analysis between p-S937-GLI1 and p-p38. **e**, **f** Cumulative survival rates of 26 MB$_{SHH}$ cases with high or low levels of cytosolic p-S937-GLI1 or p-p38. Median survival times are shown. Data were presented as mean ± sd, two-tailed Student's $t$ test for (**b**, **c**), Pearson product-moment correlation coefficient analysis for (**d**), Log-rank analysis for (**e**, **f**). Source data are provided as a Source Data file.

Gli1(S941A) knock-in mice have not developed any phenotype of MB$_{SHH}$, this is consistent with the notions that overexpression of Gli1 alone is insufficient to induce MB[43], that overexpression of Gli1 in combination of activation of other oncogenes is able to induce MB in mice[44], and that ablation of *Gli1* (*Gli1⁻/⁻*) causes normal and well-developed cerebella in mice[45,46]. Our Gli1(S941A) knock-in mice substantially give rice to BCC-like proliferation in epidermis, resembling human superficial BCC[47]. This appears to be at odds with the finding that epidermal transgene of *Gli1* is sufficient to develop other 4 types of human BCC-like neoplasia in addition to superficial BCC[47]. The discrepancy could be explained by the difference of Gli1 expression level between our result and the other study. The substantial role of Gli1(S941E) knock-in in reducing the incidence and severity of *Smo-M2*-derived MB$_{SHH}$ prompts us to speculate the essential role of Gli1 in the oncogenesis of MB$_{SHH}$ and the potentially resistant mechanism underling Smo inhibitors vismodegib and sonidegib[48]. Gli1(S941E) knock-in significantly increases the overall survival rate in *Smo-M2*-derived MB$_{SHH}$ and p-Ser937-GLI1 positively correlates to the overall survival rate of children with MB$_{SHH}$, both suggest the determinant role of phosphorylation on Ser937/SerS941 of GLI1 in the incidence and severity of MB$_{SHH}$ and BCC.

Taken together, the amplitude of SHH signaling is fine-tuned in part via the mechanism of p38α-mediated GLI1 phosphorylation on Ser937/Ser941 that substantially contribute to the incidence and severity of MB$_{SHH}$ and BCC, and intervention of this critical phosphorylation site of GLI1 may provide an additional approach for pharmacotherapy of these diseases.

## Methods
This study was approved by the Ethics Committee of the Children's Hospital of Zhejiang University School of Medicine and the Institutional Animal Care and Use Committee of Hangzhou City University School of Medicine.

### Clinical specimens and ethics approval
The medical records and the remaining paraffin-embedded sections of children with pathologically proven MB$_{SHH}$ (26) and Group3/4 MB (20) were obtained from Department of Pathology, the Children's Hospital of Zhejiang University School of Medicine. This retrospective study was approved by the Ethics Committee of the Children's Hospital of Zhejiang University School of Medicine (2023-IRB-0233-P-01).

## Mouse strains

Wild-type C57BL/6 J mice were purchased from Shanghai SLAC Laboratory Animal Co. Ltd. (Shanghai, China). *Smo-M2* and *Smo^f/f^* mice were purchased from the Jackson Laboratory (Bar Harbor, ME). *GFAP-Cre* mice were gifted from prof. Chong Liu at Department of Pathology, Zhejiang University School of Medicine. *Gli1(S941A)^+/-^* and *Gli1(S941E)^+/-^* founders were generated by CRISPR/Cas9 at Shanghai Model Organisms Center, Inc. (Shanghai, China). All animals were housed and bred at the Hangzhou City University School of Medicine Animal Care Facility according to the institutional guidelines for laboratory animals, and the protocol (No. 23004) was approved by the Institutional Animal Care and Use Committee of Hangzhou City University School of Medicine. All experimental animals were housed in a specific pathogen-free (SPF)-grade environment with room conditions of approximately 22–26 °C with 40–60% humidity, and a 12/12-h light cycles. We used age- and sex-matched littermates of both sexes, aged between 2 and 24 weeks in this study. 2-month-old mice were used for breeding and 2- or 6-month-old male and female mice were used for experimental observations. The diameter of tumors in mice allowed by the ethics committee is less than 20 mm. We ensured that each time mice were sacrificed, the diameter of cerebellar tumors did not exceed this limit.

## Cell culture and transfection

All the cell lines used in the present study were obtained from National Collection of Authenticated Cell Cultures (Shanghai, China) and authenticated using STR profiling. The human HEK293T (SCSP-502), and Daoy cells (SCSP-509) were maintained in high glucose DMEM (Gibco, 11995065) supplemented with 10% fetal bovine serum (FBS, Gibco, 10091148), 100 μ/ml penicillin and 100 μg/ml streptomycin (Biosharp, BL505A). The mouse C3H10T1/2 cells (SCSP-506) and MEF were cultured in MEM (Gibco, 11140050) supplemented with 10% FBS and 100 μ/ml penicillin and 100 μg/ml streptomycin. All the cells were maintained in a humidified incubator containing 5% $CO_2$ and 95% air at 37 °C. All the cells were periodically tested for free mycoplasma contamination via PCR. Transient transfection was performed using Lipofectamine 2000 reagent (Thermo Fisher Scientific, 11668019) as per the manufacturer's instructions.

## Plasmids

Human or mouse Gli1 full-length cDNA was cloned by PCR from 293T cell or C3H10T1/2 cell cDNA pool, and mutations were introduced by site-directed mutagenesis. MAPK11, MAPK12, MAPK13, and MAPK14 was cloned into pXJ-40-Flag and all variants of Gli1 were cloned into pXJ-40-HA vector. Sequences of MAPK and Gli1 variants were confirmed by DNA sequencing. The hairpin shRNA templates targeting Gli1 of complementary oligonucleotide containing overhangs were digested and inserted into *XbaI* and *NotI* sites of a lentiviral shRNA expression vector, PLL3.7. The shRNA target sequences: CCAGCCC AGATGAATCACCAA and ACCCAACTTGCCCAATCACAA.

## Antibodies, proteins, and chemicals

The following antibodies were used in this study: JNK (#9252, Cell Signaling Technology, 1:1000 for WB), ERK1/2 (#4695, Cell Signaling Technology, 1:1000 for WB), p38 (#8690, Cell Signaling Technology, 1:1000 for WB), phospho-JNK (#4668, Cell Signaling Technology, 1:1000 for WB), phospho-ERK1/2 (#9212, Cell Signaling Technology, 1:1000 for WB), phospho-p38 (ab178867, Abcam, 1:1000 for WB, 1:100 for IHC), α-tubulin (sc-8035, Santa Cruz, 1:1000 for WB), β-actin (sc-69879, Santa Cruz, 1:1000 for WB), K5 (sc-80606, Santa Cruz, 1:100 for IHC), K10 (sc-53251, Santa Cruz, 1:100 for IHC), K17 (sc-393002, Santa Cruz, 1:100 for IHC), GAPDH (sc-32233, Santa Cruz, 1:1000 for WB), Ser/Thr antibody (05368, Millipore, 1:1000 for WB), Gli1 (ab217326, Abcam, 1:500 for WB, 1:100 for IHC, 1:50 for IP), Gli2 (ab26056, Abcam, 1:500 for WB), Gli3 (ab69838, Abcam, 1:500 for WB), Smo (ab236465, Abcam, 1:1000

for WB), Ki67 (ab15580, Abcam, 1:100 for IHC), BLBP (ab32423, Abcam, 1:200 for IF), Histone H3 (AF0009, Beyotime Biotechnology, 1:500 for WB), Flag (AF0036, Beyotime Biotechnology, 1:1000 for WB), HA (AF2305, Beyotime Biotechnology, 1:1000 for WB, 1:50 for IP), Pax6 (ET1612, Hua-An Biotechnology, 1:100 for IF), NeuN (ET1602, Hua-An Biotechnology, 1:100 for IF), Calbindin (ET1702, Hua-An Biotechnology, 1:100 for IF), Myc (R1208, Hua-An Biotechnology, 1:1000 for WB, 1:50 for IP). anti-rabbit IgG (#7074, Cell Signaling Technology, 1:10000 for WB), anti-mouse IgG (#7076, Cell Signaling Technology, 1:10,000 for WB), Alexa-555 (S21381, Life Technology, 1:1000 for IF) and Alexa-488 (S32354, Life Technology, 1:1000 for IF) were used as secondary antibodies. Bioactive recombinant human N-SHH protein were from R&D Systems (PRD1845-01, Minneapolis, MN, USA). Cyclopamine (S1146), Purmorphamine (S3042), SP600125 (S1460), SB203580 (S1076), PD98059 (S1177), and Losmapimod (S7215) were purchased from Selleckchem (Munich, Germany). SAG (HY-12848C) and Cycloheximide (HY12320) were from MedChemExpress (New Jersey, USA), MG132 (S1748) was from Beyotime Biotechnology (Shanghai, China).

## Generation of an antibody against p-Ser937-GLI1

Rabbit anti-phospho-S937-GLI1 polyclonal antibody was raised by repeated immunization of rabbits with a purified peptide with the sequence of SQV(pS)PSRAKA (GLS Biochemistry, Shanghai, China). The synthetic peptide was conjugated to Keyhole limpet hemocyanin (KLH) and bovine serum albumin (BSA), and then were used for immunization and conjugation evaluation, respectively. The KLH-peptide conjugate was used as an antigen to immunize female New Zealand White rabbits 6 times for 7 weeks according to standard protocols. Rabbit serum was filtered through a 0.45 μm filter and the antibody was purified using an affinity chromatography column prepared by coupling immunogenic peptide to SulfoLink coupling resin (Thermo Scientific, Massachusetts, USA). The specificity of the antibody was verified by GLI1-shRNA and neutralizing polypeptide.

## Dual-luciferase reporter assays

Cells were co-transfected with an 8×Gli-luciferase reporter construct and a Renilla luciferase reporter construct in the presence or absence of interest Gli1 variants. Luciferase activities were measured using the dual luciferase reporter assay kit (Promega, Madison, WI, USA) according to the manufacturer's instruction. Renilla reporter gene expression was used as an internal standard. Data were obtained by analyzing triplicated samples each prepared from three independent experiments.

## RNA isolation and quantitative real-time PCR (qRT-PCR)

Total RNA was isolated from C3H10T1/2 cells, MEFs or cerebella by using a TRIzol reagent (Takara Biotechnology, Dalian, China) according to the manufacturer's instructions. 2 μg RNA in a volume of 20 μl was reversely transcribed by using SuperScript III reagent (Life Technologies) and the oligo-(deoxythymidine) primers were incubated at 42 °C for 1 h. After the termination of cDNA synthesis, mRNA levels of target genes were determined by qRT-PCR. The relative amounts of the mRNA levels of target genes were normalized to the GAPDH levels, respectively, and the relative difference in mRNA levels was calculated by $2^{-\Delta\Delta Ct}$ method. The indicated primers are listed in the Supplementary Table 1.

## Western blotting and immunoprecipitation experiments

Protein extraction was performed by using RIPA and protein concentrations were determined by using a standard Bradford assay. 50 μg of total protein was subjected to SDS-PAGE followed by a transfer onto PVDF membranes (Millipore, Billerica, MA, USA). Membranes were incubated with primary antibodies against Gli1, phospho-S937-Gli1, Gli2, Gli3, α-tubulin, GAPDH, β-actin, p-ERK1/2, ERK1/2, p-

JNK, JNK, p-p38, p38, Flag, Histone H3, HA, and p-Ser/Thr followed by incubation in secondary antibodies. Immunoprecipitation were performed in whole cell lysates by using various antibodies followed by SDS-PAGE and immunoblotting. Immunosignals were developed by using the Enhanced Chemiluminescence System. GAPDH and Histone H3 were used as internal standards.

## Histology, immunofluorescence, immunohistochemistry, and TUNEL staining

Tissues were isolated and fixed overnight in 4% paraformaldehyde (PFA) and embedded in paraffin. Paraffin sections were subjected to dewaxing, hydration, and antigenic repair (microvaved in sodium citrate antigen repair solution for 20 min), and blocking (5% goat serum in PBS for 1 h). For immunofluorescence, the samples were incubated with primary antibody against Pax6, NeuN, Calbindin, BLBP, and HA overnight at 4 °C, followed by incubation with Alex555- or Alex488-conjugated secondary antibody at room temperature for 1 h. Nuclei were counterstained with 6'-diamidino-2-phenylindole (DAPI), and immunostaining was analyzed by a laser confocal microscope (LSM900). For immunohistochemistry, the samples were incubated with primary antibody against p-p38, Gli1, Ki67, phospho-S937-Gli1, K5, K10 and K17, respectively, overnight at 4 °C, followed by incubation with biotin-labeled goat anti-mouse/rabbit IgG secondary antibody and 3,3'-diaminobenzidine tetrahydrochloride (DAB) developing using an immunohistochemistry kit from ZSGB-BIO (Beijing, China). TUNEL staining was performed following the manufacturer's protocol using a TUNEL staining kit (Roche, Shanghai, China), and a fluorescence microscope was utilized to examine the TUNEL-positive cells.

## Protein digestion and liquid chromatography triple quadrupole mass spectrometry (LC-MS/MS)

The mass spectrometry analysis was performed by Micrometer Biotech Company. Endogenous human GLI1 from Daoy cells was enriched by co-immunoprecipitation with anti-Gli1 and purified by SDS-PAGE. The gel was then stained with Coomassie brilliant blue, and the corresponding visualized bands were excised, destained with ammonium bicarbonate buffer and dehydrated in 75% acetonitrile. After rehydration with ammonium bicarbonate, trypsin digestion was performed, followed by peptide extraction with acetonitrile, drying, and mass spectrometry[49].

## Statistics and reproducibility

No statistical methods were used to predetermine sample size. Sample sizes were estimated based on experiences on the similar experiments performed by us and other published studies. Methods of statistical analysis employed to determine the significance of difference are reported in related figure legends, and the exact p values are indicated in figures. Statistical calculations were performed using the Prism software (GraphPad).

## Reporting summary

Further information on research design is available in the Nature Portfolio Reporting Summary linked to this article.

# Data availability

The proteomics and IP-MS publicly available data used in this study have been deposited to the ProteomeXchange Consortium via the iProX partner repository with the dataset identifier PXD038222 and PXD046374. Source data are provided with this paper. All remaining data is available in the Article, Supplementary and Source Data files. Source data are provided with this paper.

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

## Acknowledgements

This work was supported by National Natural Science Foundation of China (No. 32170841, 31871395, 31801207, 82000046, 81801396), Key R&D Program Project of Zhejiang (No.2022C03034), and Zhejiang Pro-vincial Natural Science Foundation of China (No. LQ21H010001). We thank Chong Liu from Zhejiang University School of Medicine for the *GFAP-Cre* mice.

## Author contributions

L.H.Z. contributed vital idea and contributed to the animal experiments; C.T., M.L.Y., Q.Q.H., M.Y.Q., Q.L.R., Y.N.X., T.Y.S. and X.J. mainly per-formed the experiments; W.Z.G. and C.T. collected clinical specimens; C.Y.X. and C.C.Z conducted data analysis; X.M.W. designed experiments and prepared the manuscript; J.R.W. worked on the experiments, first draft writing and revision.

## Competing interests

The authors declare no competing interests.
