## [Peer Review File · Nature Communications]

Reviewers' Comments:

Reviewer #1:

Remarks to the Author:

In this study, the authors provide evidence that p38-mediated phosphorylation of GLI1 at position S939 leads to GLI1 destabilization and suppression of Hedgehog (HH) signaling. Genetic activation of HH signaling in the context of medulloblastoma and basal cell carcinoma reduces the amount of pGLI1 and active p38 kinase, enhancing the oncogenic effect of HH signaling.

These data contribute to a deeper understanding of the complex post-translational regulatory mechanisms in the control of HH signaling and have the potential to enable clinically and therapeutically innovative approaches.

However, the reviewer is not left with an overwhelmingly enthusiastic impression because innovation, novelty value, and methodology have some shortcomings.

Major points:

1. S/T phosphorylation of GLI proteins has previously been shown in numerous studies, which the authors do not reference comprehensively. For instance, Huntzicker et al. (2006) have identified degron sequences in GLI, which if phosphorylated, lead to degradation of GLI1 and hence reduced GLI activity. In the present study, the authors add another potential phosphorylation site involved in the control of GLI stability. Therefore, the degree of innovation and novelty appears somewhat reduced.

2. The role of p38 in the regulation of HH/GLI, even in medulloblastoma, has been addressed and published before. These studies have not been referenced comprehensively, which is critical since the results show an opposite effect (e.g.: <https://doi.org/10.1007/s00401-012-0946-z> and <https://doi.org/10.1016/j.jaut.2022.102902> and <https://doi.org/10.1111/j.1471-4159.2009.05900.x>)

How can this be reconciled with the data presented in this manuscript?

3. What is the mechanism of destabilization? Does it alter the level of ubiquitination as has been shown in previous studies on GLI phosphorylation?

4. The authors claim that S939 is the critical amino acid in human, while in the functional studies using murine Gli1 they use S941 despite the presence of S939, which is more likely to be the homologous site.

5. The phosphorylation sites in GLI1 in response to HH/SAG and MAPK14/p38 activity need to be analyzed by mass-spec methodology. Using antibodies may not reveal data with sufficient quality and resolution.

6. Quantitative data of Western blots and in vivo stainings are largely missing, making the interpretation of the data difficult, particularly in light of the known lack of high-quality GLI antibodies. Original blots need to be supplied to the reviewers to better judge the results presented.

7. There are many Western blots where total and phosphor-proteins apparently have been analyzed on different lanes/membranes. Standard would be to probe total and phospho-forms sequentially or with differently labelled antibodies though using the same blotting membrane.

8. The authors show substantial reduction of p38/MAPK14 (Fig 1B) in response to SmoM2 activation. When screening the supplementary excel table, p-p38 reduction is only modest (S2/S1: 0.89). Furthermore, the reviewer could not find the data for pGLI1 S939/941 in the table. Did the authors find any pGLI1 peptides in the mass-spec phosphor-proteomics approach?

9. Figure H-J: pGLI1 staining is not very convincing and for the reviewer hard to see the differences described in text and shown in the graphs; J: high pGLI1 levels should result in degradation and lower total GLI1 levels; how about p38/MAPK14 activity status and correlation with survival? This should also be compared to non-SHH MB subtypes.

Reviewer #2:

Remarks to the Author:

In this manuscript, Wang and colleagues show that activation of HH signaling inactivates p38 (MAPK14), which, in turn, directly phosphorylates GLI1 on Ser937/Ser941 (human/mouse) site to induce its proteasomal degradation and to inhibit the transcription of HH signaling. As a result, Gli1-S941A gain-of-function knock-in in mice phenocopies the Gli1 transgene in causing robust hyperplasia of epidermis resembling human superficial BCC, and Gli1-S941E loss-of-function

knock-in significantly reduces the incidence and severity of Smoothed-M2 transgene-induced spontaneous SHH-MB. Interestingly, authors demonstrate that phospho-Ser937-Gli1, a destabilized form of Gli1, is predominantly localized in the cytosol and positively correlates to the overall survival rate of children with SHH-MB. This study uncovers Gli1 phosphorylation on Ser937/Ser941 by p38a as a novel mechanism controlling HH signaling in oncogenesis of SHH-MB and BCC, providing a target for therapeutic intervention.

The manuscript is clear and well written. Experiments are professionally executed and statistical analysis is accurate and well performed. Results are clearly presented and the main conclusions are supported by the results. Methodology is sound and there enough detail provided in the methods for the work to be reproduced.

Minor comments:

1. In the result section discussing Fig. 2K authors mention that "CHX chase assay indicated overexpression of MAPK14 significantly shortened the half-life of endogenous Gli1 protein in C3H10T1/2 cells". Although it is clear that overexpression of MAPK14 decreases Gli1 protein expression, the authors should perform densitometry on Gli1 to better support this claim and show the degree of diminished half-life of endogenous Gli1 protein in presence of MAPK14.

2. In the results discussing Fig. 2L authors state that proteasomal inhibitor MG132 markedly restored the MAPK14's effect on Gli1 protein expression. Again, it would be nice to perform densitometry on Gli1 to better support this claim.

3. The Gli-luciferase reporter assay in Figure 2B would be better controlled with a mutated GLI binding site to show specificity.

Minor points:

-Pag. 5, line 61, please correct "servers" with "serves".

-Pags. 6-7, lines 78 and 92, please correct "BBC" with "BCC".

-Pag. 9, line 138, please replace "negated" with "reduced" or a more appropriate term. "negated" is reported several times throughout the manuscript.

-Pag. 15, lines 252 and 263, please correct "Gli11(S941A)" with "Gli1(S941A)".

Reviewer #3:

Remarks to the Author:

The manuscript examines the role of Gli1 phosphorylation at serine 937 (human)/941 (mouse) in SHH pathway activation, and in the tumorigenesis of BCC and SHH-driven medulloblastoma. Starting with a phosphoproteomic screen comparing medulloblastomas that form in transgenic Smo-mutant mice to normal cerebella in age matched littermates without expressed Smo mutation, the team finds SHH activation causes decreased phosphorylation of p38, which correlated with increased Gli1. They then confirmed that decreased p38 phosphorylation was Smo-dependent and that p38 activity correlated with Gli1 phosphorylation. They mapped Gli1 phosphorylation to serine 937 using a series of site-directed Gli1 serine mutations in reporter co-transfection assays. Based on these data, the team made two mutant mouse lines, one with Gli1 S941A mutation and one with Gli1S941E mutation and then examined SHH pathway activation and developmental phenotype.

Gli1 S941A, which was predicted to stabilize Gli1, was said to cause cerebellar overgrowth and skin hyperproliferation. The evidence for cerebellar phenotype is not strong, however, as no volume comparisons, cross-sectional area comparisons or cell counts are presented. The histologic data that is presented is not quantitative and does not clearly show qualitative differences. The histologic evidence in the skin is stronger.

Gli1S941E mutation, which was predicted to destabilize Gli1 was said to also disrupt cerebellar growth and to reduce medulloblastoma tumorigenesis. The evidence for this tumor suppressive effect was stronger, with decreased numbers of Smo-mutant mice showing medulloblastoma in mice with Gli1S941E mutation, compared to Smo-mutant mice without Gli1S941E mutation. There

is some data to support the relevance of Gli1 939 phosphorylation to medulloblastoma in human patients.

Overall, the paper elegantly documents a modulation of SHH signaling by p38 through phosphorylation of Gli1. However, there are issues with the brain phenotype data that must be addressed. The interpretation of the cerebellar phenotype in mice with Gli1 S941A mutation seems overstated. The phenotype needs to be more carefully and quantitatively documented, or the conclusions that the mutation alters cerebellar development need to be scaled back or removed. Additionally, the authors should clarify if the K-M curve showing survival of Smo-mutant mice with Gli1S941E mutation includes all mice with this genotype (which I consider would be appropriate) or only mice with this genotype that developed medulloblastoma (not correct). If only mice with medulloblastoma were considered in the K-M curve, the exclusion of mice with the Smo-mutant genotype that did not develop medulloblastoma would make the curve more similar to the Smo-mutants without Gli1S941E mutation, inappropriately reducing the size of the effect. As a minor point, it is also not clear what is meant by the numbers 119 and 114 in the figure legend for Fig 4H "Overall survival rates of GFAP-Cre;Smo-M2 (119) and GFAP-Cre;Smo- M2;Gli1(S941E)+/+ (114) pups with pathologically proven MBSHH" and clarification is needed. Another minor point is that Western blot analyses are referred to incorrectly as "western analysis" and this needs to be fixed in every instance. With these changes, the evidence reasonably supports the conclusions of the manuscript.

A remaining issue is that the significance of the findings to the treatment of SHH-driven cancers is not as high as it might be. It is not clear how p38-driven Gli1 phosphorylation can be usefully targeted, since decreased phosphorylation, as might be achieved with a small molecule inhibitor is oncogenic. It would address this concern if the authors could show if p38-driven Gli1 phosphorylation modulates the efficacy of SMO inhibitor therapy in BCC or medulloblastoma. If so, tumor-driven suppression of p38 activity may be a resistance mechanism. Without this experiment, the authors need to make a stronger case for translational significance.

Response to Reviewers

Reviewer #1 - SHH, phosphoproteomics, GEMs (Remarks to the Author):

In this study, the authors provide evidence that p38-mediated phosphorylation of GLI1 at position S939 leads to GLI1 destabilization and suppression of Hedgehog (HH) signaling. Genetic activation of HH signaling in the context of medulloblastoma and basal cell carcinoma reduces the amount of pGLI1 and active p38 kinase, enhancing the oncogenic effect of HH signaling.

These data contribute to a deeper understanding of the complex post-translational regulatory mechanisms in the control of HH signaling and have the potential to enable clinically and therapeutically innovative approaches.

However, the reviewer is not left with an overwhelmingly enthusiastic impression because innovation, novelty value, and methodology have some shortcomings.

Major points:

Comment #1: S/T phosphorylation of GLI proteins has previously been shown in numerous studies, which the authors do not reference comprehensively. For instance, Huntzicker et al. (2006) have identified degron sequences in GLI, which if phosphorylated, lead to degradation of GLI1 and hence reduced GLI activity. In the present study, the authors add another potential phosphorylation site involved in the control of GLI stability. Therefore, the degree of innovation and novelty appears somewhat reduced.

Response #1: We greatly appreciate the reviewer's insightful and constructive comments that really improve the quality of manuscript. We have now cited the literature in the Discussion section (**Line 404-405**). Huntzicker et al. have identified two independent destruction signals in Gli1 (D_N and D_C) that are involved in regulating GLI1 stability, however, our identified S937 phosphorylation site of GLI1 is located outside the destruction domain of GLI1. In addition, though S/T phosphorylation of GLI proteins has previously been shown in numerous studies, we have highlighted another important phosphorylation site (Ser937) that is phosphorylated by p38 α and substantially involved in oncogenesis of MB_{SHH} and BCC as well. Most importantly, we for the first time shed light on the correlation between the p-Ser937-GLI1 level and the clinical prognosis of MB_{SHH}.

Comment #2: The role of p38 in the regulation of HH/GLI, even in medulloblastoma, has been

addressed and published before. These studies have not been referenced comprehensively, which is critical since the results show an opposite effect (e.g.: <https://doi.org/10.1007/s00401-012-0946-z> and <https://doi.org/10.1016/j.jaut.2022.102902> and <https://doi.org/10.1111/j.1471-4159.2009.05900.x>). How can this be reconciled with the data presented in this manuscript?

Response#2: We agree this is a critical issue needed to be addressed emphatically.

Guldal et al. (<https://doi.org/10.1007/s00401-012-0946-z>) revealed the increase in p-p38 α levels in proliferating cerebellar granule neuron precursors (CGNPs), in cerebellar granule cells overexpressing constitutively active form of *Smo*, and in human MB tumor tissues versus paracancerous tissues, Zhu et al. (<https://doi.org/10.1016/j.jaut.2022.102902>) showed activation of SHH signaling promoted p-p38 levels in synovial fibroblasts in RA, and Atkinson et al. (<https://doi.org/10.1111/j.1471-4159.2009.05900.x>) indicated the G-protein receptor kinase 2-dependent p38 activation regulated Shh-mediated gene transcription in cultured astrocytes.

Notably, we performed immunohistochemistry staining and western blot analyses in whole cerebella expressing constitutively active form of *Smo* and Cre recombinase under the human GFAP promoter. This GFAP-Cre recombinase appears in the rhombic lip as early as E13.5 and in the astrocytes (including Bergmann glial cells) and granular cells in addition to the Purkinje cells at postnatal day 14 (DOI: 10.1186/1756-6606-6-25). Moreover, we compared the p-p38 levels between the human MB_{SHH} and non-WNT/non-SHH MB, and treated the C3H10T1/2 cells, SHH-responsive cells, with SHH and p38 α inhibitor, losmapimod.

The discrepancy could indicate that the role of p38 in SHH signaling depends on the cellular context and the duration, degree, and/or cellular location of subtype (p38 α , p38 β , p38 γ , and p38 δ) activation. In this regard, the exact role of p38 α in SHH signaling in vivo remains to be further investigated, and it would be of interest in future studies to complete the characterization of p38 α inactivation mice and to determine the role of p38 α in cell proliferation and tumorigenesis of MB_{SHH}.

We have now cited the literatures and added these notions in the Discussion section (**Ref. 40, 41, and 43; Line 428-431, and 434-437**).

Comment #3: What is the mechanism of destabilization? Does it alter the level of ubiquitination as has been shown in previous studies on GLI phosphorylation?

Response#3: We have performed an additional experiment and reveal that overexpression of MAPK14 (P38 α) significantly enhances the ubiquitination of GLI1 in 293T cells transiently expressing Myc-GLI1 and HA-ubiquitin (**Figure 20**).

Comment #4: The authors claim that S939 is the critical amino acid in human, while in the functional studies using murine Gli1 they use S941 despite the presence of S939, which is more likely to be the homologous site.

Response#4: We apologize for the incorrect description. Since murine Gli1(S941) is homologous to human GLI1(S937) (**Figures 2C-2F**), we use murine Gli1(S941) mutant mice for functional study. We have corrected this description in the current version of manuscript (**Line 204-214**).

Comment #5: The phosphorylation sites in GLI1 in response to HH/SAG and MAPK14/p38 activity need to be analyzed by mass-spec methodology. Using antibodies may not reveal data with sufficient quality and resolution.

Response#5: We thank the reviewer for the professional comment. We have now performed an additional mass spectrometric analysis for GLI1 proteins purified from human medulloblastoma Daoy cells, which reveals six endogenous phosphorylation sites, including Ser70, Ser569, Ser927, Ser937, Thr601 and Thr1074 (**Supplementary Figure 4**).

Protein	Pos	Leading	Protein	Protein	Gene	Fasta hit	Localiza	Score d	PEP	Score	Delta sc	Score fc	C	Nt	Ami	Sequence window
P08151	937	P08151	P08151	Zinc fingr	GLI1	>sp P081	0.99873	28.967	1.07E-12	217.16	173.4	217.16	1	S	1	EPSYQSPKFLGGSQVSPSRAPVNTYGPGF
P08151	601	P08151	P08151	Zinc fingr	GLI1	>sp P081	0.67621	3.19953	0.00074	105.71	72.522	105.71	1	T	1	HYLLRARYASARGGGTPTAASLDRIIGLP
P08151	1074	P08151	P08151	Zinc fingr	GLI1	>sp P081	0.85035	10.0876	2.03E-14	170.59	163.69	71.117	1	T	1	LSPPSHDQRGSSGHTPPPSGPPNMAVGNMS
P08151	640	P08151	P08151	Zinc fingr	GLI1	>sp P081	1	76.0151	0.00132	76.015	58.12	76.015	1	S	1	EYPGYNPNAGVTRRASDPPAQAADRPAPARVQ
P08151	569	P08151	P08151	Zinc fingr	GLI1	>sp P081	0.97084	18.7239	0.00023	65.962	55.706	65.962	1	S	1	TVSRSSLASPFPPGSPPEAGASSLPGLMPA
P08151	927	P08151	P08151	Zinc fingr	GLI1	>sp P081	0.99067	20.2586	4.17E-09	198.77	179	198.77	1	S	1	GGGREDAPAQEPSYQSPKFLGGSQVSPSRAK
P08151	146	P08151	P08151	Zinc fingr	GLI1	>sp P081	0.9999	39.9594	1.17E-16	212.65	178.2	212.65	1	S	1	PSLGFFAQMNHQKGPSFVQPCGPHDSAR
P08151	70	P08151	P08151	Zinc fingr	GLI1	>sp P081	0.9009	9.58584	2.01E-13	214.18	194.68	214.18	1	S	1	PARETNSCTEGLFSSPSRAVKLTKKRALS

Comment #6: Quantitative data of Western blots and in vivo staining are largely missing, making the interpretation of the data difficult, particularly in light of the known lack of high-quality GLI antibodies. Original blots need to be supplied to the reviewers to better judge the results presented.

Response#6: We have quantified all the western and IHC results. In addition, we have uploaded all the raw images of western blots, and reviewers are invited to consult them as needed.

Comment #7: There are many Western blots where total and phosphor-proteins apparently have been analyzed on different lanes/membranes. Standard would be to probe total and phosho-forms sequentially or with differently labelled antibodies though using the same blotting membrane.

Response#7: We agree with the reviewer's comments. Ideally, total and phospho-proteins would be probed sequentially or with differently labelled antibodies in the same blotting membrane. Unfortunately, some stripped membranes are not applicable to probing total proteins after probed with the phospho-proteins. In addition, some antibodies against total proteins and against phospho-proteins have the same labelling. Due to these reasons, we used the same protein samples to analyze the total and phosphorylated proteins on different membranes under exactly the same conditions.

Comment #8: The authors show substantial reduction of p38/MAPK14 (Fig 1B) in response to SmoM2 activation. When screening the supplementary excel table, p-p38 reduction is only modest (S2/S1: 0.89). Furthermore, the reviewer could not find the data for pGLI1 S939/941 in the table. Did the authors find any pGLI1 peptides in the mass-spec phosphor-proteomics approach?

Response#8: We apologize for mis-uploading the Excel file that belonged to another MB study. We have now uploaded the quantitative iTRAQ LC-MS/MS phosphoproteomic data from *GFAP-Cre;SMO-M2* pups and *Smo-M2* control littermates (n=3) at 2 weeks of age. This data show that the p-p38 (MAPK14) reduction is approximately 0.415 (S2/S1: 0.415) (**Supplementary Table 1**). Unfortunately, we have not yet obtained any pGLI1 peptides in the mass-spec phosphor-proteomics.

Protein	Positi	Ratio	Regul	P value	Amin	Protein	descr	Gene	name	Localiza	PEP	Score	Modified	sequence	Charge	Mass	erro	MS/MS	Cov	WJR1_1	WJR1_2	WJR1_3	WJR2_1	WJR2_2	WJR2_3
P47811	2	0.415	Down	0.0013128	S	Mitogen-activ	Mapk14			0.999976	7.05E-05	106.38	S(1)QERPTFYR	2	-0.74091	5	1.211	1.43	1.418	0.502	0.698	0.48			

Comment #9: Figure 3 H-J: pGLI1 staining is not very convincing and for the reviewer hard to see the differences described in text and shown in the graphs; J: high pGLI1 levels should result in degradation and lower total GLI1 levels; how about p38/MAPK14 activity status and correlation with survival? This should also be compared to non-SHH MB subtypes.

Response#9: As recommended by the reviewer, we repeated the immunostaining and also performed additional experiments to quantify the p-GLI1(S937) and p-P38 levels. We recruited 46 children with MB, and 20 and 26 of them were pathologically diagnosed with Group 3/4 MB and MB_{SHH}, respectively. Results showed cytosolic p-GLI1(S937) levels were significantly lower in MB_{SHH} than in non-WNT/non-SHH MB (group 3/4 MB), whereas p-p38 levels were consistently reduced in MB_{SHH} versus group 3/4 MB. Moreover, correlation analysis indicated cytosolic p-GLI1 levels positively correlated with p-p38 levels in MB_{SHH} and the R² value was 0.5656 ($p < 0.0001$). Long-rank analysis showed that the overall survival of children with high cytosolic p-GLI1 expression was significantly longer than that of children with low cytosolic

p-GLI1 expression, whereas there was no significant difference in overall survival between children with high p-p38 expression and those with low p-p38 expression (**Figure 5**).

Reviewer #2 - SHH, MAPK (Remarks to the Author):

In this manuscript, Wang and colleagues show that activation of HH signaling inactivates p38 (MAPK14), which, in turn, directly phosphorylates GLI1 on Ser937/Ser941 (human/mouse) site to induce its proteasomal degradation and to inhibit the transcription of HH signaling. As a result, Gli1-S941A gain-of-function knock-in in mice phenocopies the Gli1 transgene in causing robust hyperplasia of epidermis resembling human superficial BCC, and Gli1-S941E loss-of-function knock-in significantly reduces the incidence and severity of Smoothed-M2 transgene-induced spontaneous SHH-MB. Interestingly, authors demonstrate that phospho-Ser937-GLI1, a destabilized form of GLI1, is predominantly localized in the cytosol and positively correlates to the overall survival rate of children with SHH-MB. This study uncovers GLI1 phosphorylation on Ser937/Ser941 by p38a as a novel mechanism controlling HH signaling in oncogenesis of SHH-MB and BCC, providing a target for therapeutic intervention.

The manuscript is clear and well written. Experiments are professionally executed and statistical analysis is accurate and well performed. Results are clearly presented and the main conclusions are supported by the results. Methodology is sound and there enough detail provided in the methods for the work to be reproduced.

Minor comments:

Comment #1: In the result section discussing Fig. 2K authors mention that “CHX chase assay indicated overexpression of MAPK14 significantly shortened the half-life of endogenous Gli1 protein in C3H10T1/2 cells”. Although it is clear that overexpression of MAPK14 decreases Gli1 protein expression, the authors should perform densitometry on Gli1 to better support this claim and show the degree of diminished half-life of endogenous Gli1 protein in presence of MAPK14.

Response #1: We are grateful to the reviewer for the insightful and constructive comments that help to improve the quality of manuscript, and have now performed densitometry analyses for Western blots throughout the manuscript.

Comment #2: In the results discussing Fig. 2L authors state that proteasomal inhibitor MG132 markedly restored the MAPK14's effect on Gli1 protein expression. Again, it would be nice to perform densitometry on Gli1 to better support this claim.

Response #2: We have now performed densitometry analyses throughout the manuscript.

Comment #3: The Gli-luciferase reporter assay in Figure 2B would be better controlled with a mutated GLI binding site to show specificity.

Response #3: We have now performed an additional Gli-luciferase reporter assay using the 8×3'Gli-luc (5'-GACCACCCA-3') construct and the control 8×m3'Gli-luc (5'-GAAGTGGGA-3') mutant construct, respectively (**Figure 2B**).

Comment #4:

Minor points:

-Pag. 5, line 61, please correct “servers” with “serves”.

-Pags. 6-7, lines 78 and 92, please correct “BBC” with “BCC”.

-Pag. 9, line 138, please replace “negated” with “reduced” or a more appropriate term. “negated” is reported several times throughout the manuscript.

-Pag. 15, lines 252 and 263, please correct “Gli11(S941A)” with “Gli1(S941A)”.

Response #4: We apologize for these mistakes, and have now corrected them accordingly.

Reviewer #3 - MB therapy and initiation (Remarks to the Author):

The manuscript examines the role of Gli1 phosphorylation at serine 937 (human)/941 (mouse) in SHH pathway activation, and in the tumorigenesis of BCC and SHH-driven medulloblastoma. Starting with a phosphoproteomic screen comparing medulloblastomas that form in transgenic Smo-mutant mice to normal cerebella in age matched littermates without expressed Smo mutation, the team finds SHH activation causes decreased phosphorylation of p38, which correlated with increased Gli1. They then confirmed that decreased p38 phosphorylation was Smo-dependent and that p38 activity correlated with Gli1 phosphorylation. They mapped Gli1 phosphorylation to serine 937 using a series of site-directed Gli1 serine mutations in reporter co-transfection assays. Based on these data, the team made two mutant mouse lines, one with Gli1 S941A mutation and one with Gli1S941E mutation and then examined SHH pathway activation and developmental phenotype.

Comment #1: Gli1 S941A, which was predicted to stabilize Gli1, was said to cause cerebellar overgrowth and skin hyperproliferation. The evidence for cerebellar phenotype is not strong, however, as no volume comparisons, cross-sectional area comparisons or cell counts are presented. The histologic data that is presented is not quantitative and does not clearly show qualitative differences. The histologic evidence in the skin is stronger.

Response #1: We greatly appreciate the reviewer's constructive comments that really improve the quality of manuscript. We have now qualified the cerebellar weights and lobule numbers in Gli1(S941A)^{-/-} and Gli1(S941A)^{+/+} pups. Moreover, we have now performed immunofluorescence staining of cerebellar sections to identify the morphological alteration in these cerebella by using Pax6, NeuN, and calbindin antibodies and shown no significant cerebellar developmental abnormality in Gli1(S941A)^{+/+} pups (**Figures 3E and 3F**). In addition, we have quantified the results of immunohistochemistry staining in skin sections and shown that Gli1(S941A)^{+/+} mice developed a BCC-like phenotype (**Figures 3I and 3K**).

Comment #2: Gli1S941E mutation, which was predicted to destabilize Gli1 was said to also disrupt cerebellar growth and to reduce medulloblastoma tumorigenesis. The evidence for this tumor suppressive effect was stronger, with decreased numbers of Smo-mutant mice showing medulloblastoma in mice with Gli1S941E mutation, compared to Smo-mutant mice without Gli1S941E mutation. There is some data to support the relevance of Gli1 939 phosphorylation to medulloblastoma in human patients. Overall, the paper elegantly documents a modulation of SHH signaling by p38 through phosphorylation of Gli1. However, there are issues with the brain phenotype data that must be addressed. The interpretation of the cerebellar phenotype in mice with Gli1 S941A mutation seems overstated. The phenotype needs to be more carefully and quantitatively documented, or the conclusions that the mutation alters cerebellar development need to be scaled back or removed.

Response #2: We thank the reviewer for positive comments on this work. According to the reviewer's advice, we have now quantified cerebellar weights and lobule numbers, and performed immunofluorescence staining using antibodies of calbindin, NeuN, and BLBP, markers of PCs, granule neurons, and BGs, in *GFAP-Cre;Smo-M2;Gli1(S941E)^{+/+}* cerebella at P14 (Figures 4E-4G). In addition, the interpretation of the cerebellar phenotype in pups with the Gli1(S941A) mutation has been modified in the current version of manuscript (Lines 275-285).

Comment #3: Additionally, the authors should clarify if the K-M curve showing survival of Smo-mutant mice with Gli1S941E mutation includes all mice with this genotype (which I consider would be appropriate) or only mice with this genotype that developed medulloblastoma (not correct). If only mice with medulloblastoma were considered in the K-M curve, the exclusion of mice with the Smo-mutant genotype that did not develop medulloblastoma would make the curve more similar to the Smo-mutants without Gli1S941E mutation, inappropriately reducing the size of the effect.

Response #3: According to the reviewer’s comments, we have now re-calculated the K-M curve by using showing *GFAP-Cre;Smo-M2* pups with MB (119) and *GFAP-Cre;Smo-M2;Gli1(S941E)^{+/+}* pups (212) with or without pathologically proven MB (Figure 4L; Line 342-349).

Comment #4: As a minor point, it is also not clear what is meant by the numbers 119 and 114 in the figure legend for Fig 4H “Overall survival rates of GFAP-Cre;Smo-M2 (119) and GFAP-Cre;Smo-M2;Gli1(S941E)^{+/+} (114) pups with pathologically proven MB_{SHH}” and clarification is needed.

Response #4: We have now changed the description accordingly (**Figure 4L legend; Line 342-349**).

Comment #5: Another minor point is that Western blot analyses are referred to incorrectly as "western analysis" and this needs to be fixed in every instance. With these changes, the evidence reasonably supports the conclusions of the manuscript.

Response #5: We have now corrected them accordingly.

Comment #6: A remaining issue is that the significance of the findings to the treatment of SHH-driven cancers is not as high as it might be. It is not clear how p38-driven Gli1 phosphorylation can be usefully targeted, since decreased phosphorylation, as might be achieved with a small molecule inhibitor is oncogenic. It would address this concern if the authors could show if p38-driven Gli1 phosphorylation modulates the efficacy of SMO inhibitor therapy in BCC or medulloblastoma. If so, tumor-driven suppression of p38 activity may be a resistance mechanism. Without this experiment, the authors need to make a stronger case for translational significance.

Response #6: We agree with the reviewer's comments and have now performed additional in vitro experiments to clarify the role of P38 activity in regulating the chemotherapeutic efficacy of SMO inhibitors. Our additional data do indicate p38 α -mediated phosphorylation on Ser937 of GLI1 was involved in the proliferation, migration, and invasion of Daoy medulloblastoma cells and in the regulation of the sensitivity to chemotherapy of SMO antagonist, cyclopamine (**Supplementary Figures 6A-6D; Line 350-363**).

Reviewers' Comments:

Reviewer #1:

Remarks to the Author:

The reviewer would like to thank the authors for the comprehensive revision and clear presentation of the changes and additions.

The authors have addressed the major concerns and integrated additional data on clinical relevance and molecular mechanisms of post-translational regulation of the GLI1 oncogene into the study. This has significantly improved the quality and relevance of the paper.

Reviewer #2:

Remarks to the Author:

The authors have added a significant amount of additional information and data to support their original claims. The revised manuscript addresses all of my concerns.

Reviewer #3:

Remarks to the Author:

The authors have thoughtfully and effectively addressed all of my concerns. I consider that the work is now ready for publication.